# A Principled Approach to Randomized Selection under Uncertainty: Applications to Peer Review and Grant Funding

**Alexander Goldberg**  **Giulia Fanti**  **Nihar B. Shah**

Carnegie Mellon University
{akgoldbe,gfanti,nihars}@andrew.cmu.edu

## Abstract

Many decision-making processes involve evaluating and selecting items, including scientific peer review, job hiring, school admissions, and investment decisions. These domains feature error-prone evaluations and uncertainty about outcomes, which undermine deterministic selection rules. Consequently, randomized selection mechanisms are gaining traction. However, current randomized approaches are ad hoc and, as we prove, inappropriate for their purported objectives. We propose a principled framework for randomized decision-making based on interval estimates of item quality. We introduce MERIT (Maximin Efficient Randomized Interval Top-$k$), which maximizes the worst-case expected number of top candidates selected under uncertainty represented by overlapping intervals. MERIT provides optimal resource allocation under an interpretable robustness notion. We develop a polynomial-time, practically efficient algorithm and prove our approach satisfies desirable axiomatic properties not guaranteed by existing methods. Experiments on synthetic peer review data from grant funding and conferences demonstrate that MERIT matches existing algorithms' expected utility under fully probabilistic models while outperforming them under our worst-case formulation.

## 1 Introduction

In many applications like scientific funding, job hiring, school admissions, and startup investment, decision makers evaluate and select items based on imperfect assessments. Recently, there has been growing interest in introducing randomization into selection processes to address uncertainty in evaluations, reduce reviewer burden, encourage high-risk proposals, and combat reviewer partiality. In fact, many funding agencies have already adopted partial lotteries to allocate grant money, starting with the New Zealand Health Research Council in 2013 [31], followed by the Swiss NSF in 2019 [1], and recently expanding to numerous agencies worldwide [14, 50, 58]. Proposals for randomization have also emerged in college admissions [1, 25], job screening [40, 8] and startup investment [35].

In current deployments, decision makers collect peer-review assessments and run a lottery where selection probabilities derive from review scores. However, current procedures are ad hoc. We initiate a principled approach to randomizing decisions from evaluations, focusing on the question: *Given imperfect evaluations of candidate quality, what is a suitable probability distribution over applicants for random selection?*

We take the perspective of a funder selecting the highest quality grant proposals. A key motivation for randomization is uncertainty about relative quality. Many existing deployments describe peer review lotteries as random "tie-breaking" between proposals of equal quality [36, 53]. Similar to

39th Conference on Neural Information Processing Systems (NeurIPS 2025).

previous work [21], we assume the funder estimates numeric quality *intervals* for each proposal. These intervals may capture uncertainty due to evaluation errors, miscalibration, subjectivity, or aleatoric uncertainty about future success. Crucially, we capture settings with *"Knightian uncertainty,"* where a funder cannot assign a probability measure over possible outcomes [26]—a model widely applied in policy-making [52, 45, 7], financial investment [13, 33, 37], and R&D investment [3].

This assumption is particularly apt for peer review, where probabilistic models of reviewer errors have performed poorly in real deployments [27, 51, Section 'Miscalibration'], decision makers lack ground truth data to evaluate model appropriateness, and future success is inherently difficult to predict [49, 12, 60, 29]. Without a reliable probabilistic model, we assume decision makers describe uncertainty through quality intervals rather than point estimates. The funder draws conclusions only about relative ordering: overlapping intervals indicate insufficient evidence to conclude one proposal is better, while non-overlapping intervals indicate dominance. We develop theory and a practical algorithm for making funding decisions from interval quality estimates, with the following **key contributions**:

1. *Modeling uncertainty as Knightian uncertainty intervals:* Prior work assumes known probabilistic relationships between quality and scores [21]. We show that in such fully Bayesian settings, deterministic selection always maximizes expected utility, making randomization unnecessary. Our model captures the motivation for randomizing using "Knightian" uncertainty intervals.

2. *A principled approach to randomization:* We formalize two key principles: *ex ante optimality*—maximizing worst-case utility over all rankings consistent with intervals—and *ex post validity*—respecting strict dominance relationships. Prior heuristic rules fail to satisfy both principles.

3. *Efficient algorithm*: We develop a polynomial time algorithm solving the maximin optimization problem, despite related graph problems being NP-hard [15, 42, 61, 2]. Our **M**aximin **E**fficient **R**andomized **I**nterval **T**op-$k$ (**MERIT**) algorithm runs in under 5 minutes on 10,000+ candidates on a standard laptop. Implementation available at `github.com/akgoldberg/lottery`.

4. *Axiomatic comparison*: We initiate an axiomatic approach to comparing randomized mechanisms, identifying desirable properties of "monotonicity in budget", "stability", and "reversal symmetry." We prove MERIT prevents "maximal instability" and respects "reversal symmetry" while existing mechanisms do not.

5. *Empirical comparison:* We evaluate MERIT against existing methods using synthetic data based on real peer review data from conferences (NeurIPS 2024, ICLR 2025) and grant agencies (Swiss NSF 2020). MERIT performs comparably to existing methods in expected utility under a linear reviewer error model used by the Swiss NSF [21] and many other prior works [16, 5, 43, 44]. However, under our worst-case objective, our algorithm significantly outperforms deterministic selection and the Swiss NSF's randomized approach.

For clarity of presentation, all formal proofs are deferred to Appendix A.

## 2 Background and Approach

We begin by describing current deployments of randomized decisions in scientific funding and then motivate our approach.

### 2.1 Existing Deployments

In recent years, there have been many deployments of "peer review lotteries" in scientific funding decisions. Most deployments use an approach of *"randomize-above-threshold."* Under this approach, the funder chooses a minimum acceptable quality threshold and samples uniformly at random among all proposals that are above this threshold. This approach has been adopted by numerous funding agencies [14, 55, 58, 50, 31] Council, the British Academy [55] and as a means of allocating oral presentations at the USENIX Security Conference [57]. However, as we describe in Section 3, randomize-above-threshold may violate a desired principle of "ex post validity", which says that if one proposal clearly dominates another, the stronger proposal should be funded if the weaker proposal is funded.

Taking a different approach, the *Swiss National Science Foundation (NSF)* [21, 56] pioneered a method that explicitly accounts for uncertainty about the quality of each proposal. They assume that

each proposal has a latent true quality and review scores are generated based on these quality scores and reviewer-specific noise parameters. They assume priors on the model parameters and develop methods to obtain point estimates and confidence intervals for the true quality of each proposal. As described in Algorithm 8, the Swiss NSF then samples $k$ proposals by setting a "provisional funding line" as the $k$-th highest point estimate. All proposals with intervals strictly above the funding line are selected and all proposals strictly below the funding line are rejected. The remaining budget is allocated uniformly at random among proposals with intervals that overlap the funding line. We provide pseudocode for the Swiss NSF method and randomize-above-threshold in Appendix E.

The intuition provided by the Swiss NSF for their approach is that confidence intervals capture the funder's uncertainty in estimating the proposal quality and randomizing decisions accounts for this uncertainty, thereby leading to better decisions. In contrast, as we prove in Theorem A.1, when the funder assumes that review data is generated from a fully specified Bayesian model, there exists a deterministic selection of proposals that maximizes the funder's expected utility for any utility function. Informally, *if the funder knows the model that generates their data, then they do not need to randomize in order to maximize their expected utility.* We present the formal proposition and proof in Appendix A.1. This result suggests one drawback of the Swiss NSF's method—in their model setting there may be a utility cost to randomization compared to choosing deterministically in an optimal manner. A second drawback is that the Swiss NSF's algorithm for selecting proposals from intervals violates natural axioms for a selection rule, like monotonicity in the budget $k$ and stability, as we show in Section 5.

## 2.2 Our Approach

In our work, we propose a model that captures the motivation for randomizing due to uncertainty about the relative quality of the proposals. We show that if the funder cares about their *worst case utility* they must randomize decisions in order to robustly optimize their utility.

Specifically, we consider a funder who estimates intervals for each proposal based on data. These intervals need not come from any one particular model, but they should capture the funder's inherent uncertainty about the relative quality of proposals.

Our interpretation of the quality intervals stems from the intuitive argument that if the intervals for two proposals overlap then the funder does not have enough evidence to distinguish between them. On the other hand, if the interval of proposal $A$ dominates the interval of proposal $B$, then the funder has sufficient evidence to believe that $A$ is better than $B$. Hence, the intervals define a partial ordering of proposals that represents a set of conclusions by the funder regarding the relative quality of different proposals. This ordering is the canonical "interval order" for a set of intervals. It captures the spirit of how the Swiss NSF interprets confidence intervals in their setup citeheyard2022rethinking.

In practice, such intervals can arise from a variety of sources, including missing-data imputation, model ensembling, expert input, multi-criteria aggregation, or robustness to model misspecification for confidence intervals. We provide detailed explanations of such intervals in Appendix F.

## 3   Problem Formulation

Our method applies to settings like admissions, scientific peer review, job screening, and financial investment, where decision makers estimate quality intervals and select top candidates based on these intervals. For concreteness, throughout our exposition, we will describe a *funder* choosing *proposals*.

Consider a funder who receives $n$ proposals. From these, the funder wishes to select the $k$ highest quality proposals. Note that $n$ could be as large as thousands of proposals and $k$ a fixed fraction of the total and can also be in the hundreds or thousands. For each proposal $i \in [n]$[1], the funder estimates an interval $[\ell_i, u_i] \subseteq \mathbb{R}$ representing a range of quality scores that the proposal could possibly take. A higher score indicates higher quality. The funder wishes to design a randomized selection mechanism to choose $k$ proposals given the intervals. In order to design such a mechanism, we adopt two primary principles which we term as ex ante optimality and ex post validity, described below.

---

[1] We use the standard notation $[\kappa]$ to denote set $\{1, \ldots, \kappa\}$ for any positive integer $\kappa$.

**Ex ante optimality** The funder's utility is the expected number of the true top-$k$ proposals that they select, ranked by their quality. Formally, let $\sigma : [n] \to [n]$ denote a ranking of the proposals where for each proposal $i \in [n]$, the rank of the proposal is denoted by $\sigma(i) \in [n]$. If the funder samples proposals with marginal probabilities $p \in [0, 1]^n$ and the true ranking of proposals is $\sigma$, their expected utility is $\sum_{i=1}^n p_i \mathbb{1}\{\sigma(i) \leq k\}$. Clearly, if the funder knew the true ranking $\sigma$, then they would optimize utility by choosing deterministically, i.e., by setting $p_i = 1$ for $i$ with $\sigma(i) \leq k$, and 0 otherwise.

However, recall that the funder is uncertain about relative qualities of the proposals, as captured by overlaps in intervals. Any ordering that is consistent with overlapping intervals could be the true ranking. Hence, the intervals define a set of feasible rankings:

$$\Sigma_n = \{\sigma \text{ permutation of } [n] \mid \forall i, j \in [n], \; \ell_i > u_j \implies \sigma(i) < \sigma(j)\}$$

In other words, if proposal $i$ has quality strictly above proposal $j$, then $i$ is ranked higher than $j$ in all $\sigma \in \Sigma_n$. As an example of an extreme case, if all $n$ intervals overlap each other, then $\Sigma_n$ consists of all possible permutations of the proposals.

For ex ante optimality, the funder optimizes their *worst case* utility over feasible rankings $\Sigma_n$:

$$\min_{\sigma \in \Sigma_n} \sum_{i=1}^n p_i \mathbb{1}\{\sigma(i) \leq k\} \tag{1}$$

The funder maximizes their worst-case expected utility by choosing the optimal marginal probabilities $p$ solving the maximin optimization problem:

$$\max_{\substack{p \in [0,1]^n: \\ \|p\|_1 = k}} \min_{\sigma \in \Sigma_n} \sum_{i=1}^n p_i \mathbb{1}\{\sigma(i) \leq k\}. \tag{2}$$

Finally, the funder randomly chooses $n$ proposals with marginal probabilities corresponding to $p$.

The ex ante optimization problem has a game theoretic interpretation that motivates the need for randomization. Our model corresponds to a zero-sum Stackelberg game where the funder is the "leader" who selects $k$ proposals. The funder faces an adversarial "follower" who chooses a ranking of proposals. The leader's utility is the number of top $k$ proposals selected based on the adversary's ranking, while the adversary's utility is the negation of the leader's. The Strong Stackelberg Equilibrium (SSE) is exactly the solution to Objective 2. It is well known that in an SSE, the leader may need to commit to a randomized (or mixed) strategy.

**Ex post validity** The ex post validity criterion requires that for any pair of proposals $a$ and $b$, if $b$'s quality interval lies strictly below $a$'s interval and if $b$ is selected, then $a$ must also be selected.

Formally, a selection rule that takes as input a set of quality intervals $I$ and outputs a set of selected proposals $S$ satisfies *ex post validity*, if for all pairs of intervals $a, b \in I$ with $\ell_a > u_b$, and all outputs $S$ selected with non-zero probability, $b \in S \implies a \in S$.

The ex post validity criteria ensures that the actual selected set of proposals is legitimate to stakeholders. In particular, if the funder rejects a proposal that dominates an accepted proposal, that would be unacceptable to the funder and to applicants.

While the ex post condition seems natural, the simple randomize-above-threshold mechanism can violate it: Suppose proposals $a$ and $b$ both lie above the threshold, but $a$ dominates $b$. Because $a$ and $b$ are entered into a uniform lottery, $a$ may be rejected at random, while $b$ is accepted at random.

# 4  Efficient Algorithm

The ex ante optimization problem in (2) is equivalent to solving the following linear program (LP):

$$\max_{p\in\mathbb{R}^n, v\in\mathbb{R}} \quad v \tag{3}$$

$$\text{subject to} \quad v \le \sum_{i=1}^{n} p_i \mathbb{1}\{\sigma(i) \le k\}, \quad \forall \sigma \in \Sigma_n,$$

$$\sum_{i=1}^{n} p_i = k \text{ and } 0 \le p_i \le 1, \forall i \in [n]$$

This LP has exponentially many constraints ($\binom{n}{k}$). The minimum weight $k$-ideal problem on arbitrary partial orders is NP-hard [15], and several related interval order problems are also NP-hard [2, 42, 61]. Despite this, we show the problem is solvable in polynomial time using the ellipsoid method with a separation oracle (Section 4.1). We develop a practical cutting plane algorithm (Section 4.2) and show how to ensure ex post validity efficiently (Section 4.3). We call this end-to-end algorithm **M**aximin **E**fficient **R**andomized **I**nterval **T**op-$k$ (**MERIT**).

## 4.1  Polynomial Time Algorithm

We now develop a polynomial-time algorithm to solve linear program (3). Our approach solves the problem using a polynomial time "separation oracle" [18] with the ellipsoid algorithm. A separation oracle checks whether a proposed solution satisfies all the constraints. If the solution is feasible, the oracle confirms it. If not, it identifies (at least one) specific constraint that the solution violates. The separation oracle may be used to solve the LP without enumerating all (exponentially many) constraints by starting with a limited set of constraints and iteratively shrinking the possible feasible region of the LP through calls to the separation oracle. Our main result proves that this method yields a polynomial time algorithm:

**Theorem 4.1** (Polynomial time solution). *The linear program* (3) *can be solved within accuracy $\epsilon$ of the optimal solution in polynomial time with respect to $n$ and $\log(1/\epsilon)$ using the ellipsoid algorithm with Algorithm 1 as a separation oracle.*

---

**Algorithm 1** Polynomial-time Separation Oracle

---

**Input:** Candidate solution: $(p, v) \in [0,1]^n \times \mathbb{R}$ with $\|p\|_1 = k$, number selected $k$, set of intervals $\{\ell_i, u_i\}_{i \in [n]}$ sorted in decreasing order of lower bound $\ell_i$
**Output:** A set of violated constraints ($\emptyset$ if $(p, v)$ is feasible)
1: $Z \leftarrow \emptyset$
2: **for** $i = 1$ to $k+1$ **do**
3:      $S_i \leftarrow \{j \in (i, n] : \text{intervals } j \text{ and } i \text{ overlap}\}$
4:      **if** $|S_i| \ge (k - (i-1))$ **then**
5:          Obtain $\tilde{S}_i$ by sorting $S_i$ by $p$ and keeping only the $k - (i-1)$ smallest values
6:          **if** $v > \sum_{j=1}^{i-1} p_j + \sum_{j \in \tilde{S}_i} p_j$ †
7:              $Z \leftarrow Z \cup \left\{ \text{``} v \le \sum_{j=1}^{i-1} p_j + \sum_{j \in \tilde{S}_i} p_j \text{''} \right\}$ **then**
8: **return** $Z$
    †*By convention, we take the empty sum from $j = 1$ to $0$ to be $0$.*

---

The primary technical difficulty is the design of an efficient separation oracle. We present our separation oracle in Algorithm 1. Given a candidate solution $(p, v)$, the separation oracle checks whether $p$ achieves a worst-case objective value of at least $v$. If the worst-case objective value under $p$ is greater than $v$, then the solution is feasible, if not the oracle returns a set of violated constraints. At a high level, the oracle works by constructing worst-case possible sets of top-$k$ proposals. For each of the $k + 1$ intervals with the largest lower bounds, the algorithm constructs the worst-case set of top-$k$ proposals that includes intervals 1 to $(i - 1)$ and excludes interval $i$ in the top-$k$. Excluding an interval with a large lower bound constrains the set of intervals that must be in the top-$k$, since all intervals strictly below the interval with the large lower bound must be excluded from the top-$k$.

In considering all such sets of intervals, the algorithm enumerates possible worst-case permutations with respect to $p$ in time $O(nk)$. If the separation oracle finds a permutation that gives objective values smaller than $v$, it returns this permutation, which corresponds to a violated constraint in the LP. If it does not find any such permutation, then $(p, v)$ is feasible. This separation oracle is used as a sub-routine in the ellipsoid algorithm to compute the optimal solution in polynomial time in Theorem 4.1.

**Lemma 4.2** (Polynomial-time separation oracle). *For any candidate solution $(p, v)$ to the linear program* (3)*, Algorithm 1 returns $\emptyset$ only if the candidate solution is feasible and returns a non-empty set of violated constraints otherwise. Moreover, Algorithm 1 runs in time $O(n \max\{k, \log n\})$.*

## 4.2 Practical Algorithm

The cutting plane algorithm is described in full in Algorithm 2. The algorithm starts by solving a relaxation of the LP without any of the worst-case value constraints to find an initial (potentially infeasible) candidate solution $(p, v)$. Then, the algorithm repeatedly calls the separation oracle to check the feasibility of the current candidate solution. If the candidate solution is feasible, it is an optimal solution to the LP, since it is optimal for a relaxation of the full LP. If the candidate solution is infeasible, the algorithm adds the constraints returned by the separation oracle and re-solves the LP.

The cutting plane algorithm converges to a feasible optimal solution quickly in practice because it is initialized with a useful set of constraints on the feasible region of the problem. These constraints prune the problem and impose monotonicity and symmetry constraints on the marginal probabilities $p$, based on the number of intervals above and below each proposal, which we define below.

**Definition 4.3** (Number above ($A$) and below ($B$)). For each proposal $i$: $A(i) = |\{r : \ell_r > u_i\}|$ and $B(i) = |\{r : \ell_i > u_r\}|$.

**Definition 4.4** (Monotonically ordered subset). Subset $M \subseteq [n]$ is *monotonically ordered* if $\forall i \in [|M| - 1]$, $A(M[i]) \leq A(M[i + 1])$ and $B(M[i]) \geq B(M[i + 1])$.

Full analysis is in Appendix B.

---

**Algorithm 2** Cutting Plane Algorithm

---

**Input:** Number of proposals $k$, intervals $I = \{\ell_i, u_i\}_{i \in [n]}$, max iterations $T$
**Output:** Ex ante optimal vector $p$
*# Prune Intervals*
1: For intervals strictly below $\geq k$ others, set $p_i = 0$ and remove.
2: For intervals strictly above $\geq n - k$ others, set $p_i = 1$ and remove.
3: Let $a$ = # accepted intervals. Update $k \leftarrow k - a$.
*# Initialize Linear Program*
4: Compute $A(i)$ = # proposals strictly above $i$, $B(i)$ = # proposals strictly below $i$.
5: Using $A, B$, partition intervals into $w$ monotone subsets $M_1, \ldots, M_w$ (Alg. 5).
6: Solve LP to obtain initial $p, v$:

$$\min_{v,p} v \quad \text{s.t.} \quad \sum_{i=1}^{n} p_i = k, \; p_i \in [0, 1] \; \forall i, \; v \leq \sum_{j=1}^{k} p_j,$$

$$p_{M[i]} \geq p_{M[i+1]} \; \forall i \in [|M| - 1], M \in \{M_1, \ldots, M_w\}$$

*# Add Cuts*
7: **for** $T$ iterations **do**
8:     $C \leftarrow \texttt{SeparationOracle}((p, v), k, I)$
9:     **if** $C = \emptyset$ **then return** p         ▷ Feasible
10:     **else** Add constraints from $C$ to LP and resolve for new $(p, v)$   ▷ Infeasible
11: **return** Failure

---

## 4.3 Enforcing Ex Post Validity

A solution to the ex ante optimality LP (3) returned by the Cutting Plane Algorithm (Algorithm 2) or the Ellipsoid Algorithm, is not guaranteed to output a vector of marginal probabilities, such that sampling proposals with these marginals always guarantees ex post validity. However, we prove that

we can post-process any solution to the ex ante optimization problem, and then sample with marginal probabilities $p$ to guarantee ex ante optimality and ex post validity simultaneously. This stands in contrast to the commonly used "randomize-above-threshold" approach to randomization, which does not guarantee ex post validity as described in Section 3.

**Theorem 4.5** (Post-processing for ex post validity). *Given any ex ante optimal $p$, Algorithm 3 enables the funder to sample $k$ proposals while satisfying both ex ante and ex post conditions and is computable in time $O(n^2)$.*

Theorem 4.5 applies the post-processing algorithm given in Algorithm 3 to a solution from the Cutting Plane Algorithm. For any $a, b \in [n]$ with $\ell_a > u_b$, Algorithm 3 terminates with $p_a = 1$ or $p_b = 0$. Moreover, Algorithm 3 never decreases the objective value of $p$. Hence, applying post-processing to an ex ante optimal solution is without loss of optimality and ensures that any sampling method that selects proposals with marginal probabilities $p$ satisfies ex post validity.

We then implement the sampling step using systematic sampling [32] which has runtime $O(n)$ (described in Appendix C).

---

**Algorithm 3** Post-Processing of $p$ for Ex Post Validity

---

    **Input:** Vector of marginal probabilities $p$, sequence of intervals $\{[\ell_i, u_i]\}_{i \in [n]}$
    **Output:** Vector of marginal probabilities $p$
1: Order the intervals by increasing $u$.
2: **for** $b \in [n]$ **do**
3:     **if** $p_b = 0$ **then**
4:         continue
5:     **for** $a$ from $n$ to $(b+1)$ **do**
6:         **if** $\ell_a > u_b$ and $p_a < 1$ **then**
7:             $d \leftarrow \min\{p_b, 1 - p_a\}$
8:             $p_b \leftarrow p_b - d$
9:             $p_a \leftarrow p_a + d$

---

### 4.4 Full Algorithm

The complete MERIT algorithm solves the ex ante optimization with post-processing for ex post validity (Algorithm 3) followed by sampling. The algorithm is provably polynomial time using the ellipsoid method (Theorem 4.1). In practice, we use the cutting plane algorithm (Algorithm 2), which is efficient albeit with non-polynomial theoretical convergence.

---

**Algorithm 4** MERIT Algorithm

---

    **Input:** Number of proposals to select $k$, set of intervals $I = \{\ell_i, u_i\}_{i \in [n]}$
    **Output:** Selection of $k$ proposals
1: Compute an ex ante optimal vector of marginal probabilities $p$ using Algorithm 2.
2: Apply ex post validity post-processing to $p$ (Algorithm 3).
3: Sample $k$ proposals from $[n]$ with marginal probabilities of inclusion given by $p$ (Algorithm 6).

---

## 5 Axiomatic Comparison

In applications like scientific funding or college admissions, there is no agreed upon ground-truth measurement of selection quality. Hence, it is unclear how to empirically measure whether one algorithm performs better than another. Therefore, inspired by social choice theory [10], we initiate an axiomatic comparison of MERIT with alternative methods from Section 2.1—deterministic top-$k$ selection, randomize above threshold, and the Swiss NSF method. We analyze the behavior of MERIT and alternatives with respect to natural axioms.

### 5.1 Defining Axioms

We propose three natural desiderata for algorithms selecting proposals from quality assessments. We begin by defining a generic "randomized selection rule."

**Definition 5.1** (Selection rule). A *selection rule* receives as input $n$ quality estimates $I = \{(\ell_i, e_i, u_i)\}_{i \in [n]}$ where $\ell_i, u_i$ are lower and upper limits on item $i$'s quality and $e_i \in [\ell_i, u_i]$ is a point estimate. Given budget $k \in \{1, \dots, n\}$, it outputs a subset of $[n]$ of size $k$. Let $p(I, k) \in [0, 1]^n$ denote the marginal selection probabilities.

This captures methods that do not use intervals (deterministic selection), use only intervals (MERIT), and use both intervals and point estimates (Swiss NSF). Next, we define three axioms.

First, **"monotonicity in budget"** requires that increasing $k$ should not decrease any proposal's selection probability:

**Definition 5.2** (Monotonicity in budget). A selection rule respects monotonicity in budget if for any input $I$ and all budgets $k \in [n-1]$, $p(I, k+1)_i \geq p(I, k)_i \; \forall i \in [n]$.

Second, a selection rule should be "stable"—changing one interval should not drastically change algorithm behavior. We define an undesirable form of instability. An algorithm exhibits **"maximal instability"** if changing a single interval by an arbitrarily small amount can switch behavior between deterministic selection (minimum entropy) and uniform random sampling (maximum entropy):

**Definition 5.3** (Maximum instability). A selection rule is *maximally unstable* if there exist inputs $I$ and $J$ differing by arbitrarily small $\epsilon > 0$ in one proposal's quality estimate and budget $k \in \{2, \dots, n-2\}$ such that $p(I, k) = \frac{k}{n}\mathbf{1}_n$ (uniform random sampling on $I$) whereas $p(J, k) \in \{0, 1\}^n$ (deterministic on $J$).

A selection algorithm should *avoid* maximum instability. We restrict budget to $\{2, \dots, k-2\}$ since stability with respect to changing a single proposal is not meaningful when choosing only one proposal to accept or reject.

Finally, inspired by **"reversal symmetry"** from social choice theory [46], when selecting 1 of 2 proposals, if the quality scale is reversed (all intervals flipped), the selection rule should flip the selection probabilities:

**Definition 5.4** (Reversal symmetry). For input $I = \{(\ell_i, e_i, u_i)\}_{i \in [n]}$ where $\ell_i, e_i, u_i \in [0, 1] \forall i$, let $I^{(R)} = \{(1 - u_i, 1 - e_i, 1 - \ell_i)\}_{i \in [n]}$ be the reversed input. A selection rule selecting $k = 1$ of $n = 2$ proposals respects reversal symmetry if for any flipped inputs $I$ and $I^{(R)}$, $p(I, 1) = (p_1, p_2)$ and $p(I^{(R)}, 1) = (p_2, p_1)$.

## 5.2 Theoretical Analysis

Deterministic top-$k$ selection meets these criteria but does not account for uncertainty (violating our ex ante requirement). We characterize randomized mechanisms in Theorem 5.5:

**Theorem 5.5** (Axiomatic analysis). *Existing randomized algorithms have the following properties:*

*(a) Swiss NSF and randomize-above-threshold both exhibit maximum instability, while MERIT is never maximally unstable.*

*(b) Swiss NSF, randomize-above-threshold and MERIT all violate monotonicity in budget.*

*(c) It is not possible to simultaneously satisfy ex ante optimality and monotonicity in budget.*

*(d) Swiss NSF and randomize-above threshold violate reversal symmetry, while MERIT satisfies reversal symmetry.*

We formally prove Theorem 5.5 in Appendix 5.5. While all randomized selection rules considered vioalte monotonicity, a funder could enforce monotonicity in budget for MERIT by solving a sequence of optimization problems from 1 to $k$, but this may come at loss of ex ante optimality, as we describe in Appendix D.

# 6 Experimental Comparison of Methods

We evaluate MERIT using real peer review data from the Swiss NSF grant reviews [21], NeurIPS 2024 conference papers, and ICLR 2025 submissions. We compare performance under both (1) expected utility in a probabilistic model of reviewer behavior and (2) our worst-case utility objective. We provide additional ablations and qualitative insights into different lottery outcomes in Appendix G.

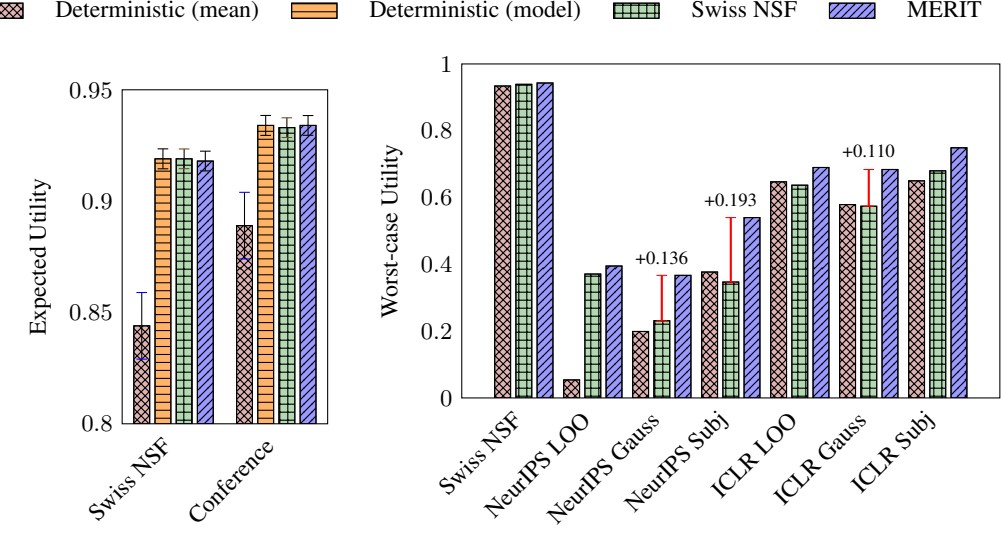

(a) Linear miscalibration model.    (b) Worst-case over interval ordering model (our model).

Figure 1: Proportion of top-$k$ proposals selected by different methods with quality data generated under the Swiss NSF model of linear miscalibration and under our model of worst-case over feasible rankings. MERIT matches performance of algorithms designed for the Swiss NSF's linear model, with expected utility averaged over 50 samples of synthetic data and error bars showing 95% CI for the sample mean. The gap between MERIT and other methods in the worst-case over intervals defined by our model can be substantial, as shown by the gaps in NeurIPS Gaussian, NeurIPS Subjectivity, and ICLR Subjectivity.

**Expected Utility under Linear Miscalibration**  We evaluate expected utility under the Swiss NSF's widely-adopted linear miscalibration model [21, 16, 5], where reviewer scores are linear combinations of true quality, reviewer bias, and Gaussian noise. We simulate two settings: (1) Swiss NSF grants (350 proposals, 10 reviewers, 80 proposals/reviewer) and (2) CS conference review (1000 papers, 1000 reviewers, 5 papers/reviewer), selecting the top one-third in both cases. We generate 50 synthetic datasets with parameters $\sigma_\theta = 2$, $\sigma_b = 1$, $\sigma_\epsilon = 0.5$ matching prior work [54], and estimate 50% confidence intervals following Swiss NSF methodology.

**Worst-Case Utility**  Our worst-case objective (1) maximizes the minimum expected fraction of true top-$k$ proposals over all rankings consistent with score intervals. We evaluate on three real datasets: Swiss NSF 2020 grants ($n = 353$), NeurIPS 2024 accepted papers ($n = 4035$), and ICLR 2025 submissions ($n = 11520$). For Swiss NSF, we use their linear model intervals. For NeurIPS and ICLR, we generate intervals using three methods: (1) leave-one-out (LOO) ranges, (2) Bayesian credibility intervals under a Gaussian model, and (3) subjectivity-adjusted intervals based on reviewer emphasis on different criteria [28, 38]. Full details are in Appendix F.

**Results**  Figure 1 shows our main results. Under the Swiss NSF's Bayesian model, MERIT performs comparably to Swiss NSF's randomized method and deterministic model-based selection in expectation, while deterministic mean-based selection performs poorly. In the worst-case setting, MERIT substantially outperforms all baselines across all datasets, achieving up to 0.19 higher utility than Swiss NSF. The optimality gap is particularly large when intervals are wide (e.g., NeurIPS LOO, NeurIPS Gaussian), where deterministic selection and even Swiss NSF's randomized approach perform poorly. These results suggest MERIT maintains expected utility under probabilistic models while providing superior worst-case robustness. We provide additional ablations and experiments in Appendix G with similar results.

**Computational Efficiency**  All experiments run on a standard 2019 MacBook Pro. Using Gurobi 12.0.1 [20] to solve LPs, MERIT completes in under five minutes even for 11,000+ proposals. Detailed runtime analysis is in Appendix H.

# 7 Related Work

**Designing Better Evaluation Processes**   Many works diagnose flaws in peer review—such as miscalibration and arbitrariness in opinions of reviewers—and propose improvements [51]. Many models assume linear miscalibration [16, 5, 43, 44, 48, 12], including those used in the Swiss NSF [21], but these often perform poorly in practice [27], likely due to the complexity of real-world miscalibration [11]. A recent approach [59] addresses arbitrary miscalibration and shows that randomized estimators can improve ranking accuracy. Related work in hiring proposes algorithmic solutions to prejudice and to uncertainty about unknown options [24, 30]. Both approaches make stronger assumptions on a known model of errors in hiring process than our work.

**Selection Under Uncertainty and Robust Optimization**   Our method falls under the umbrella of distributionally robust optimization (DRO) [41]. Our main contribution is a formulation specific to scenarios where decision makers want to select $k$ high quality items under ambiguity sets defined by intervals. Though similar in spirit to robust portfolio optimization [62], our assumptions differ as investors typically have probabilistic models and seek risk-averse diversification. Recent work on conformal prediction for selection [22] focuses on bounding false positives above a fixed threshold, whereas we select the top-$k$ items without assuming a fixed quality threshold or reliable probabilities.

**Randomized Social Choice**   Our work connects to the literature on randomized social choice, which examines how incorporating randomness into voting rules can enhance desired properties [10, 9]. Unlike our setting, most social choice models assume no underlying ground-truth quality. A related line of work in social choice theory on "distortion" interprets voters' rankings as noisy reflections of latent utilities and seeks aggregation rules that perform well under this uncertainty [4]. Our approach is conceptually similar: we consider rankings consistent with intervals of quality and design a randomized mechanism that maximizes the worst-case expected utility of the selected top-$k$ items.

# 8 Discussion

We introduce MERIT, a computationally efficient lottery for top-$k$ selection under uncertainty. By relying solely on intervals rather than fully specified generative models, MERIT respects real-world funder constraints while providing a principled robust optimization solution. Our case studies demonstrate scalability to tens of thousands of candidates. An additional benefit of MERIT is that it can handle additional constraints on the form of the lottery. For example, in some cases, funders may prefer to implement a *uniform lottery*, where all candidates subject to randomization are selected with equal probabilities. The optimization problem used in MERIT can be modified to constrain the lottery to uniform sampling, thereby implementing the ex ante optimal uniform lottery that respects ex post validity constraint (discussed in Appendix I).

**Limitations.**   MERIT exploits only interval *ordering*, appropriate when credible probabilistic models are unavailable but potentially sub-optimal with well-calibrated predictive models. We assume fixed proposal costs; variable budgets require accounting for both quality and budget allocation. Additional pairwise quality information (e.g., from common reviewers) is not currently exploited but could improve performance. Finally, MERIT may be less interpretable than threshold-based methods like Swiss NSF.

**Future work.**   Several extensions merit investigation. First, *variable-cost candidates*: allowing partial funding or incorporating cost-quality trade-offs in budget-constrained settings, with attention to incentive compatibility. Second, *additional ordering information*: developing efficient heuristics for general partial orders beyond interval orders, perhaps via additional monotonicity constraints. Third, *richer utility functions*: extending beyond 0-1 utility to positional scoring rules (e.g., Borda count) while maintaining tractability. Fourth, *cost–quality trade-offs*: analyzing efficient trade-offs between reviewer resources and decision quality to guide when and how to randomize. Finally, *equilibrium effects*: understanding how randomization alters applicant and reviewer incentives and strategic behavior. Progress on these questions can inform both theoretical understanding and practical implementation of randomized selection mechanisms.

Progress on these questions can inform both theoretical understanding and practical implementation of randomized selection mechanisms.

## Acknowledgments

This work was supported in parts by NSF 1942124 and ONR N000142212181, as well as the Gates Foundation. The views and opinions expressed in this study are those of the authors and do not necessarily reflect the views or positions of the sponsors.

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

# A Proofs

## A.1 Optimality of deterministic selection in a fully Bayesian setting (Section 2.1)

Here we provide the formal statement and proof of our informal claim that a funder can maximize their utility via a deterministic selection if they assume a fully Bayesian model of their data.

We prove a general statement for a funder who estimates a total ranking of proposals and receives a pre-specified utility for any possible estimate of the ranking of proposals and true qualities of the proposals. The top-$k$ selection problem is a case of this estimation problem where the funder's utility specifies the utility of choosing $k$ proposals based on the estimated ranking of proposals.

**Proposition A.1** (Optimality of deterministic selection in the fully Bayesian setting). *Consider a funder who estimates a ranking of proposals from review data in the following fully Bayesian setting. The funder observes review data $y \in Y$ for a set of proposals of true quality $\theta \in X$. The review data and true quality are generated jointly from a known model. Letting $\Pi$ denote the set of all permutations of proposals, the funder estimates a ranking $\hat{\pi}$ of proposals and gains utility $u(\hat{\pi}, \theta)$ where $u : \Pi \times X \to \mathbb{R}$. The funder aims to choose a (potentially randomized) estimator of the true ranking $f : Y \to \Delta(\Pi)$ that maximizes their expected utility $\mathbb{E}_{y,\theta,\hat{\pi}}[u(\hat{\pi}, \theta)]$ where $\hat{\pi} \sim f(y)$ . In this setting, there always exists a deterministic $f$ that maximizes the funder's expected utility.*

*Proof.* Expand the expected utility by conditioning on the observed data $y$ as $\mathbb{E}_{y,\theta,\hat{\pi}}[u(\hat{\pi}, \theta)] = \mathbb{E}_y[\mathbb{E}_{\hat{\pi},\theta|y}[u(\hat{\pi}, \theta)]]$. For a given $y$, the funder maximizes the inner expectation by choosing any $\hat{\pi} \in \arg\max_{\pi' \in \Pi} \mathbb{E}_{\theta|y}[u(\pi', \theta)|y]$, where the maximum over $\Pi$ always exists since the set of permutations $\Pi$ is finite. Hence, the deterministic estimator that takes $f(y) \in \arg\max_{\pi' \in \Pi} \mathbb{E}_{\theta|y}[u(\pi', \theta)|y]$ for all $y \in Y$ maximizes the funder's expected utility. $\square$

## A.2 Polynomial time algorithm for ex ante optimization (Theorem 4.1)

In order to prove the main theorem, we first prove Lemma 4.2, which establishes that the separation oracle (Algorithm 1) is correct and efficient.

*Proof of Lemma 4.2.* Let $v'$ be the smallest $v_i$ found by the separation oracle. We begin by showing that:

$$v' = \min_{\sigma \in \Sigma_n} \sum_{i=1}^{n} p_i \mathbb{1}\{\sigma(i) \leq k\}.$$

For $i \in [n]$, let

$$K = \{T \subset [n], |T| = k : \exists \sigma \in \Sigma_n \text{ s.t. } \sigma(j) \leq k \ \forall j \in S\}$$

be the set of all feasible selections of top-$k$ intervals given $\Sigma_n$. Note that we can rewrite the objective as

$$\min_{\sigma \in \Sigma_n} \sum_{i=1}^{n} p_i \mathbb{1}\{\sigma(i) \leq k\} = \min_{T \in K} \sum_{j \in T} p_j.$$

Now, we divide $K$ into subsets, for $i \in [k+1]$, define

$$K^{(i)} = \{T \subseteq K : i \notin T \text{ and } [i-1] \subset T\}$$

so that $K^{(i)}$ contains all sets in $K$ that include intervals 1 to $(i-1)$ and exclude interval $i$ and $K = \cup_{i=1}^{k+1} K^{(i)}$. Let the intervals be sorted in decreasing order of lower bound $\ell$. Then, for any $i \in [n]$, $i$ is not strictly above any of the intervals from 1 to $(i-1)$. Additionally, letting

$$S_i = \{j \in (i, n] : \text{intervals } j \text{ and } i \text{ overlap}\}$$

any interval in $(i, n] \setminus S_i$ must be strictly below interval $i$. Hence, any set of top-$k$ intervals $T \in K^{(i)}$ contains intervals 1 to $(i-1)$ and the remaining $k - (i-1)$ intervals come from $S_i$. Additionally, for all $j \in S_i$, $u_j \geq \ell_i$, while $\ell_j \leq \ell_i$ so all intervals in $S_i$ overlap each other and any selection of these $S_i$ in the top $k$ is feasible. Hence:

$$K^{(i)} = \{[i-1] \cup A \ : \ A \subseteq S_i, |A| = (k - (i-1))\}.$$

Finally, for fixed $p$, the minimum objective value over $K^{(i)}$ is to select the $(k - (i-1))$ proposals with the smallest values of $p$. Algorithm 1 exactly enumerates these worst-case constraints for each $K^{(i)}$ and hence returns the minimum over $T \in K$.

Now, the separation oracle adds constraints to $Z$ when $v > \sum_{j=1}^{i-1} p_j + \sum_{j \in S_i} p_j$. Hence the separation oracle returns $\emptyset$ only if $v \leq \sum_{j=1}^{i-1} p_j + \sum_{j \in S_i} p_j$ for every $i$, which is precisely the condition for feasibility. Further, as argued above, the separation oracle considers feasible top-$k$, so any constraints added by the separation oracle represent valid constraints for the LP.

The algorithm runs in time $O(n \max\{k, \log n\})$, because finding $S_i$ requires at most a linear scan over $O(n)$ proposals for each of $(k)$ proposals and initial sorting takes time $O(n \log n)$. We can sort intervals by $p$ once, to ensure that each linear scan for $S_i$ returns the intervals with smallest $p$ with no additional sorting per iteration. $\qquad\square$

Now, we prove the main theorem by showing that the ellipsoid algorithm converges in polynomial time using this separation oracle as a sub-routine:

*Proof of Theorem 4.1.* We may relax the equality constraint of LP (3) $\sum_{i=1}^{n} p_i = k$ to an inequality $\sum_{i=1}^{n} p_i \leq k$ without loss of optimality, since increasing any $p_i$ (up to 1) cannot decrease the worst-case value of $v$. This relaxation ensures that the feasible region lies within a full-dimensional affine subspace of $\mathbb{R}^{n+1}$, and contains a nontrivial interior. In particular, the point $p_i = \frac{k}{n}, v = 0$ lies strictly inside the box constraints and satisfies all inequalities strictly, implying the feasible region is full-dimensional.

Then, because our separation oracle runs in time polynomial in $n$, the classical ellipsoid algorithm [23] using the separation oracle to make cuts, solves the optimization problem to within accuracy $\epsilon$ in time $\text{poly}(n, \log(1/\epsilon), \log(R/r))$ where $R$ is the radius of a Euclidean ball that contains the feasible

region and $r$ is the radius of a Euclidean ball entirely contained in the feasible region. Clearly, $R$ is upper bounded by poly$(n)$, since each $p_i$ is bounded in $[0, 1]$ and $v$ is bounded in $[0, k]$. To establish a lower bound on $r$, we invoke Theorem 6.2.2 in Grötschel, Lovász, and Schrijver [19], which states that if a polyhedron $P = x \in \mathbb{R}^n : Ax \leq b$ is full-dimensional, and the matrix $A$ and vector $b$ consist of integers of maximum bit length $U$, then $P$ contains a ball of radius at least $2^{-\text{poly}(n,U)}$. Noting that all constraints in our model have integral coefficients of bit length at most $\log(k) < n$, we have that $r \geq 2^{-\text{poly}(n)}$. Hence, $\log(R/r)$ is poly$(n)$ and so the runtime of the algorithm is polynomial in $n$ and $\log(1/\epsilon)$. $\qquad \square$

### A.3 Ex post validity (Theorem 4.5)

*Proof of Theorem 4.5.* We will prove that for any $a, b \in [n]$ such that $\ell_a > u_b$ and for any input vector of marginal probabilities $p$, the post-processing Algorithm 3 terminates with $p_a = 1$ or $p_b = 0$ and never decreases the objective value of $p$. Then, if any ex ante optimal $p$ is given as input to the algorithm, after post-processing it is still ex ante optimal. Moreover, any sampling method that respects the post-processed marginal probabilities will satisfy ex post validity, since if $a$ dominates $b$, either $a$ is always sampled ($p_a = 1$) or $b$ is never sampled ($p_b = 0$).

Let $a, b$ be any intervals with $\ell_a > u_b$. Let $D = \{d \in [n] \mid \ell_d > u_a\}$ be the set of all intervals strictly above interval $a$. Note that (1) all $d \in D$ are also strictly above $b$ and that (2) $b$ precedes $a$ when ordered by $u$ and $a$ precedes all $d \in D$. Hence, Algorithm 3 will process $b$, then $a$, then $d \in D$. After processing $b = b$, either $p_b = 0$ or $p_d = 1$ for all $d \in D$. Because $p_b$ cannot increase in any subsequent iterations, if $p_b = 0$, it will remain 0 until the algorithm terminates. If $p_d = 1$ for all $d \in D$, then because $a \in D$, $p_a = 1$. Furthermore, since $p_d = 1$ for all $d \in D$, $p_a$ will not decrease in any subsequent iterations. Hence, the algorithm ends with $p_b = 0$ or $p_a = 1$.

Further, for any $\sigma \in \Sigma_n$, if $\sigma(b) \leq k$, then $\sigma(a) < \sigma(b) \leq k$. Hence, moving probability mass from $p_b$ to $p_a$ cannot decrease $\min_{\sigma \in \Sigma_n} \sum_{i=1}^n p_i \mathbb{1}\{\sigma(i) \leq k\}$. $\qquad \square$

### A.4 Axiomatic comparison (Theorem 5.5)

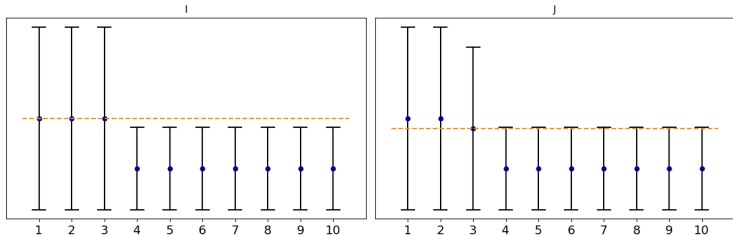

Figure 2: Example that violates *maximum instability* for Swiss NSF and randomize-above-threshold, with $k = 3$. Slightly decreasing the upper bound and point estimate of proposal 3, changes the algorithm's behavior from selecting the top 3 deterministically (left) to sampling among all 10 proposals uniformly at random (right).

**(a) Maximum instability** We will show that both Swiss NSF and randomize above threshold (with data-dependent threshold) are maximally unstable using an example. Let intervals 1 to $k$ be $[0, 2]$ with point estimates of 1 and let intervals $k + 1$ to $n$ be $[0, 1 - \epsilon]$ for some $\epsilon \in (0, 1)$. Then, the Swiss NSF algorithm selects the first $k$ proposals deterministically. Now, shift interval $k$ to be $[0, 2 - 2\epsilon]$ with point estimate of $1 - \epsilon$. Since all the intervals contain the $k$-the point estimate the Swiss NSF algorithm selects uniformly at random. Note that $\epsilon$ can be taken to be arbitrarily small, so the Swiss NSF is maximally unstable for any $\epsilon > 0$. If the randomize above threshold method chooses the threshold in a data-dependent manner, then the same example leads to maximal instability if the threshold is taken to be the $k$-th highest point estimate.

Now, to show that MERIT is not maximally unstable, we first observe that MERIT selects among all proposals uniformly at random *only if* all intervals intersect each other. Let $1 < k < n - 1$. Assume for the sake of contradiction that MERIT samples uniformly at random from all intervals and that there are two intervals $i$ and $j$ with $i$ strictly above $j$. Since the algorithm is ex post valid, if $p_j > 0$, then $p_i = 1$, but then $p_i = p_j = 1$, which is possible only if $n = k$, yielding a contradiction.

Now, to show that MERIT is *not* maximally unstable, we establish that for any set of intervals where all intervals except for one overlap, the algorithm never chooses deterministically. Since all intervals but one overlap, the non-overlapping interval can either have the largest lower bound or the smallest upper bound.

First, consider the case where interval $j$ has the largest lower bound. Then, we can partition the intervals into 3 sets $\{j\}$, $X$, the set of intervals that intersect $j$, and $B$ the set of intervals strictly below $j$. By symmetry and monotonic chain constraints (detailed in Appendix B), the algorithm will output marginal probabilities with at most 3 different values $p_j, p_x, p_b$ such that $p_j \geq p_x \geq p_b$. Since $|X| + |B| > k$, the only input for which the algorithm could output $k$ 1's is when $|X| = k - 1$ and $|B| = n - k > 1$. In this case, $j$ is always in the top $k$, so $p_j = 1$ and $j$ is pruned from the problem. Then, our algorithm chooses $k - 1$ out of $n - 1$ intervals that all intersect, so the optimal solution is sampling $k - 1$ uniformly at random from the remaining $X \cup B$ intervals. Hence, our algorithm will not choose deterministically.

Now, consider the case where interval $j$ has the smallest upper bound. Then, we can partition the intervals into 3 sets $\{j\}$, $X$, the set of intervals that intersect $j$, and $A$, the set of intervals strictly above $j$. By symmetry and monotonic chain constraints, our algorithm will output marginal probabilities with at most 3 different values $p_j, p_x, p_a$ such that $p_a \geq p_x \geq p_j$. Since $|A| + |X| \geq k + 1$, the only possible input for which the algorithm could output $k$ 1's is if $|A| = k$ and $|X| = n - k - 1 > 0$. In this case, $j$ is never in the top $k$, so the feasible top $k$ could be any subset of size $k$ from $A \cup X$. Therefore, selecting $k$ proposals uniformly at random from $A \cup X$ is optimal, so MERIT will not choose deterministically. □

**(b) + (c) Monotonicity in budget** We show that Swiss NSF, randomize above threshold, and MERIT are not monotonic in budget $k$, via the example shown in Figure 3.

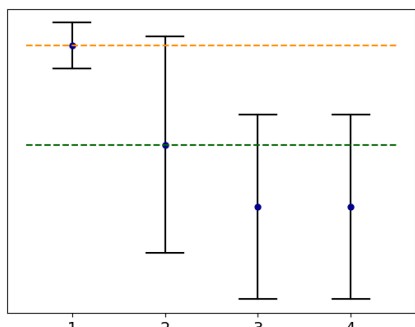

Figure 3: Example that violates monotonicity with respect to $k$ for Swiss NSF and our MERIT algorithm. When $k = 1$, $p_2 = 1/2$ for both algorithms. However, when $k = 2$, $p_2 = 1/3$ for both algorithms.

In this example, taking $k = 1$, both the Swiss NSF method and MERIT randomize between proposals 1 and 2, so proposal 2 is selected with probability $1/2$. However, if the number of proposals selected is increased to $k = 2$, then proposal 1 is always selected and both algorithms sample uniformly at random from intervals 2 to 4, meaning that proposal 2 has a selection probability of only $1/3$. Hence, even though more proposals are being selected, proposal 2 is worse-off. The same example shows that randomize-above-threshold violates monotonicity, taking the threshold to be the point estimate or the lower bound of the $k$-th highest proposal.

This example also proves that it is not possible to simultaneously satisfy ex ante optimality and monotonicity in budget. For $k = 1$, the unique ex ante optimal $p$ is $(1/2, 1/2, 0, 0)$ since feasible top 1 are $\{1\}$ and $\{2\}$. For $k = 2$, the unique ex ante optimal $p$ is $(1, 1/3, 1/3, 1/3)$ since the feasible top 2 are $\{1, 2\}$ $\{1, 3\}$, $\{1, 4\}$. Since these are unique ex ante optimal solutions, there is no sequence of solutions for $k$ equal to 1 and 2 that satisfy monotonicity in budget $k$ and ex ante optimality simultaneously.

**(d) Reversal symmetry** To prove that Swiss NSF and (data-dependent) randomize-above-threshold violate reversal symmetry, consider intervals $(0, 1)$ and $(0.1, 0.2)$ with point estimates $0.5$ and $0.15$.

Then, the Swiss NSF selection rule or randomize-above-threshold with threshold set as the highest point estimate, will accept interval 1 and reject interval 2. However, if intervals are flipped so interval 1 stays the same, but proposal 2 has interval $(0.8, 0.9)$ with point estimate of $0.85$, then Swiss NSF samples uniformly at random between the two proposals. Now, when $n = 2$ and $k = 1$, MERIT only samples between the two proposals if their intervals overlap. If the intervals are horizontally flipped, then they still overlap and MERIT samples uniformly at random respecting reversal symmetry. If interval 1 lies strictly above interval 2, then after flipping interval 2 lies strictly above interval 1, so the marginal probabilities of selection are $(1, 0)$ and $(0, 1)$ respectively, respecting reversal symmetry.

## B  Full Cutting Plane Algorithm

As described in Section 4.2, we find that the cutting plane algorithm converges to a feasible optimal solution much faster when it is initialized with additional useful constraints. These constraints prune decision variables and impose monotonicity and symmetry constraints on marginal probabilities $p$, based on the number of intervals above and below each proposal.

The cutting plane algorithm is described in full in Algorithm 2. The algorithm prunes intervals and initializing the LP with monotonicity constraints. The remainder of this section proves that these additions are without loss of optimality and ensure faster convergence.

We can think of $A, B$ as defining a partial order: $i \succeq j \iff A(i) \leq A(j)$ and $B(i) \geq B(j)$. Then, a monotonically ordered subset of intervals is a totally ordered subset or a "*chain*" and the minimum number of monotonically ordered subsets of intervals that covers a set of intervals is the "*width*" of the partially ordered set of intervals. Note that this partial order defined by $A$ and $B$ is *not* the canonical interval order that we use to define the set of feasible permutations $\Sigma_n$. The $(A, B)$ partial order never was width larger than the interval order and often has much smaller width. At the extreme, if all intervals overlap, then the width of the interval partial order is $n$, while the width of the $(A, B)$ partial order is 1, since all proposals have equal $A$ and $B$.

As we now prove, the cutting plane algorithm converges in $O(k^{w+1})$ iterations, where $w$ denotes the number of monotonically ordered subsets partitioning the intervals (or the width of the $(A, B)$ partial order). In theory, $w$ can grow linearly with $k$, so this algorithm does not have the theoretical polynomial-time guarantee of the ellipsoid algorithm. In practice, $w$ is often small for many sets of intervals.

**Proposition B.1** (Cutting plane algorithm convergence). *Letting $w$ denote the number of monotonically ordered subsets (chains) partitioning the intervals per Definition 4.4, Algorithm 2 converges to an optimal solution in $O(k^{w+1})$ iterations. The algorithm solves an LP with at most $O(n + k^{w+1})$ constraints.*

*Proof.* We first prove three lemmas that show that we can impose initial pruning, symmetry, and monotonicity constraints on $p$ without loss of optimality.

**Lemma B.2** (Pruning of optimal $p$). *There exists an optimal $p$ in which $p_i = 1, \forall i \in [n]$ with $B(i) \geq n - k$ and $p_j = 0, \forall j \in [n]$ with $A(j) \geq k$.*

*Proof.* If $B(i) \geq n - k$, then $i$ is always included in the top $k$ in any permutation of intervals. If $p_i < 1$, then there must be some other interval with $p_j > 0$, but setting $p_i = 1$, $p_j = 0$ will not decrease the objective value, since $i$ is always in the top $k$. Similarly, if $A(i) \geq k$, then $i$ is never included in the top $k$, so taking $p_i = 0$ by shifting probability mass from $i$ to any other proposal cannot hurt the objective value. $\square$

**Lemma B.3** (Symmetry of optimal $p$). *There exists an optimal $p$ in which $p_i = p_j$ for all $i, j \in [n]$ such that $A(i) = A(j)$ and $B(i) = B(j)$.*

*Proof.* If $A(i) = A(j)$ and $B(i) = B(j)$, then $i$ and $j$ have the same sets of intervals that are strictly above and strictly below each interval. Hence, for any permutation $\sigma \in \Sigma_n$ the permutation $\sigma'$ with $i$ and $j$ exchanged is also in $\Sigma_n$, so the objective value is maximized at $p_i = p_j$. $\square$

**Lemma B.4** (Ordering of optimal $p$ by $A$ and $B$). *Let $M_1, \cdots, M_w$ be a partitioning of $[n]$ such that each $M_i$ is monotonically ordered. Then, there exists an optimal $p$ for Objective (2) such that within each $M$, $p_{M[j]} \geq p_{M[j+1]}$ for all $j \in [|M| - 1]$.*

*Proof.* Let $i, j \in [n]$ be any pair of proposals with $A(i) \leq A(j)$ and $B(i) \geq B(j)$. Let $p$ be a feasible solution to the optimization problem with $p_j > p_i$. We will show that exchanging the values of $p_j$ and $p_i$ can never decrease the objective value. Define $q$ as equivalent to $p$ with $i$ and $j$ exchanged:

$$q_r = \begin{cases} p_j & r = i \\ p_i & r = j \\ p_r & \text{otherwise.} \end{cases}$$

Let $v(p, \sigma) = \sum_{i=1}^n p_i \mathbb{1}\{\sigma(i) \leq k\}$. We want to show that $\min_{\sigma \in \Sigma_n} v(p, \sigma) \leq \min_{\sigma \in \Sigma_n} v(q, \sigma)$. Consider any $\sigma \in \Sigma_n$. If $\sigma(i) < \sigma(j)$, then $v(p, \sigma) \leq v(q, \sigma)$. If $\sigma(i) > \sigma(j)$, define permutation $\tau$ to be equivalent to $\sigma$ but with $i$ and $j$ exchanged:

$$\tau(r) = \begin{cases} \sigma(j) & r = i \\ \sigma(i) & r = j \\ \sigma(r) & \text{otherwise.} \end{cases}$$

Note that $v(p, \sigma) \geq v(q, \sigma)$, but $v(q, \tau) = v(p, \sigma) \geq v(p, \tau)$. We now proposition that $\tau \in \Sigma_n$. Let $r \in [n]$ be any proposal such that $\sigma(r) \in [\sigma(j), \sigma(i)]$. Then, $r$ cannot be strictly above $j$, but since $A(j) \geq A(i)$, $r$ cannot be strictly above $i$. Similarly, $r$ cannot be strictly below $i$, but since $B(i) \geq B(j)$, $r$ cannot be strictly below $j$. Hence, $r$ must overlap both $i$ and $j$. Therefore, exchanging $i$ and $j$ does not violate any constraints, so $\tau \in \Sigma_n$. Hence, we conclude that $\min_{\sigma \in \Sigma_n} v(p, \sigma) \leq \min_{\sigma \in \Sigma_n} v(q, \sigma)$. Now, applying this exchange to every sequence of intervals within each $M_i$ yields the desired result. $\square$

Further, using symmetry and pruning together reduces the number of decision variables to $O(k)$ from $O(n)$.

**Lemma B.5** (Pruning and symmetry give $O(k)$ decision variables). *After pruning per Proposition B.2 and grouping symmetric intervals per Proposition B.3, the optimization problem has $< 2k$ decision variables.*

*Proof.* After pruning all proposals with $A(i) \geq k$ or $B(i) \geq n - k$, there are at most $k - 1$ proposals with $B(i) > 0$. The remaining intervals all have $B(i) = 0$ and can have $k - 1$ possible values of $A(i)$, so they can be grouped into at most $k - 1$ groups with equivalent $p_i$. Hence, the intervals can be grouped into at most $2k - 2$ decision variables. $\square$

Now, to complete the proof of correctness, observe that by Lemmas B.2 and B.4 the initial pruning and monotonicity constraints are without loss of optimality. We can additionally impose equality constraints on $p$ using symmetry (Lemma B.3) to reduce the dimension of $p$ to $O(k)$, although this is not shown in Algorithm 2 for simpler presentation. Because the linear program solved by the cutting plane algorithm is a relaxation of the linear program (3), a solution to the problem upper bounds the objective value of (3). By the correctness of the separation oracle (Theorem 4.2), if the cutting plane algorithm converges to a feasible $(p, v)$, this therefore is an optimal solution to the full LP.

To prove that the algorithm is guaranteed to converge within $k^{w+1}$ iterations observe that there are at most $O(k^w)$ possible total orders of marginal probabilities (after applying symmetry and pruning) consistent with the partial order given by monotonicity constraints $p_{M[i]} \geq p_{M[i+1]} \quad \forall i \in [|M|-1], \forall M \in \{M_1, \cdots, M_w\}$. Note that for any total order of the proposals, there are $k+1$ possible constraints that the separation oracle can return, because the possible constraints are determined by the order of the $p_i$. Hence, there are at most $O(k^{w+1})$ cuts that could be added to the linear program and so the cutting plane algorithm must converge in $O(S^{w+1})$ iterations. The initial LP only contains $O(n)$ constraints, so the LP never has more than $O(n + k^{w+1})$ constraints. $\square$

## B.1 Finding Minimal Set of Chains for $(A, B)$ Partial Order

In order to partition intervals using $A$ and $B$ as per Proposition B.4, we need to compute such a partition, known as a chain covering of the set. We would like the partition into as few sets as possible in order to add as many constraints as possible to the problem and reduce runtime of our optimization algorithm. There are practical general methods to solve this chain cover problem for any partial order in time $O(n^{2.5})$ by computing the maximum matching of an appropriately constructed bipartite

graph. In our case, where the partial order has specific structure, we can solve the minimal chain cover problem even more efficiently in time $O(n \log n)$. The algorithm described below is equivalent to an algorithm given in [17][Chapter 7, Algorithm 7.1] for minimal coloring of a permutation graph. For completeness, we reproduce the algorithm and proof of optimality in our problem setting below.

---

**Algorithm 5** Greedy minimal chain cover for the product order on $\mathbb{R}^2$

---

**Require:** A set of points $\{(a_i, b_i)\}_{i=1}^n$ with partial order $i \succeq j \iff a_i \geq a_j$ and $b_i \geq b_j$.
**Ensure:** A set $\mathcal{C} = \{C_1, \ldots, C_v\}$ of non-increasing chains that covers $[n]$.
1: Sort the indices in decreasing order of $a$ so that $a_{i_1} \geq a_{i_2} \geq \cdots \geq a_{i_n}$, breaking ties by decreasing $b$
2: $\mathcal{C} \leftarrow \varnothing$
3: **for** $t \leftarrow 1$ **to** $n$ **do**
4:     Let $C$ be the chain with the smallest $b_{\text{tail}(C)}$ such that $b_{\text{tail}(C)} \geq b_{i_t}$
5:     **if** $C$ exists **then**
6:         Append $i_t$ to end of $C$
7:     **else**
8:         Create a new chain $C_{\text{new}} \leftarrow \{i_t\}$ and append it to $\mathcal{C}$
9: **return** $\mathcal{C}$

---

**Proposition B.6** (Correctness of minimal chain cover algorithm). *Algorithm 5 returns a minimal size chain cover $\mathcal{C} = \{C_1, \ldots, C_v\}$ of the partially ordered set $([n], \succeq)$ defined by $i \succeq j \iff a_i \geq a_j$ and $b_i \geq b_j$. It runs in time $O(n \log n)$.*

*Proof.* First, we proposition that each $C_r$ is a chain. Indices are processed in non-increasing $a$-order, so within any chain the $a$-coordinates never increase. A point $p$ is appended to $C_r$ only if the current tail $q$ of $C_r$ satisfies $b_q \geq b_p$; hence $(a_q, b_q) \succeq (a_p, b_p)$.

Now, to show that this set of chains is minimal we will invoke Dilworth's Theorem, which states that the minimal number of chains that cover a poset (its width) is equivalent to the length of the longest antichain (sequence of incomparable elements.) We argue that the set of final tails of each chain forms an antichain. Let $T = \{t_1, \ldots, t_v\}$ be the final tails ordered so that $b_{t_1} < b_{t_2} < \cdots < b_{t_k}$. For any chains $i, j$ with $i < j$ we have $a_{t_i} \geq a_{t_j}$, otherwise $t_j$ would have been added to chain $i$. But, $b_{t_i} < b_{t_j}$, so $t_i$ and $t_j$ are incomparable. Thus the elements of $T$ are pairwise incomparable and $T$ is an antichain of size $v$. Now, let $A \subseteq [n]$ be any antichain. Map each $x \in A$ to the chain that contains it. Two distinct elements of $A$ cannot lie in the same chain, hence $|A| \leq v$. Therefore, the length of the longest antichain is $v$, so the width of the partially ordered set is $v$ and the algorithm has returned a minimal number of chains.

The algorithm can be implemented in time $O(n \log n)$, because the set of chains is always sorted in increasing order of the $b$ value of their tails so we use binary search to find the chain in which to insert each $i$ (step 4.) $\qquad \square$

## C    Systematic Sampling

The method known as "Systematic Sampling", works by first computing cumulative probabilities $S_i = \sum_{j=1}^{i} p_j$. Then, it selects a random starting point $u$ uniformly from the interval $[0,1)$ and picks exactly one item from each of the $k$ intervals obtained by adding integers $m = 0, 1, \ldots, k-1$ to the starting point $u$. Each item is selected if at least one of these evenly spaced points falls within its corresponding cumulative interval $[S_{i-1}, S_i)$. Thus, the algorithm guarantees selecting exactly $k$ distinct items without replacement, each with the correct marginal probability $p_i$. Additionally, we can initially shuffle the items uniformly at random, so that the algorithm replicates the expected behavior of uniform random sampling for items with equal values of $p$. The algorithm requires two passes over all the proposals and hence runs in time $O(n)$.

---

**Algorithm 6** Systematic Sampling [32]

---

**Require:** Integer $k$, probability vector $p \in [0,1]^n$ with $\sum p_i = k$
**Ensure:** A subset of $k$ items sampled without replacement from $[n]$ where item $i \in [n]$ is included with marginal probability $p_i$.
 1: Compute cumulative sums: $S_0 \leftarrow 0$, $S_i \leftarrow \sum_{j=1}^{i} p_j$ for $i = 1$ to $n$
 2: Sample $u \sim \text{Uniform}(0,1)$
 3: **for** each $m$ from 0 to $k - 1$ **do**
 4:     Include item $i$ in the sample where $(u + m) \in [S_{i-1}, S_i)$

---

## D  Enforcing Monotonicity in Budget

In this section, we show that it is possible to solve MERIT sequentially and simultaneously satisfy ex post validity and monotonicity in budget $k$. This may come at the cost of ex ante optimality.

---

**Algorithm 7** Solving Sequence of Optimization Problems for Monotonicity

---

**Require:** Sequence of intervals $\{[\ell_i, u_i]\}_{i \in [n]}$, number selected $k$.
 1: Let $p^{(0)} = (0, \cdots, 0)$
 2: **for** $i \in [k]$ **do**
 3:     Compute ex ante optimal $p$ for selecting top-$i$ proposals with the additional constraint $p_j^{(i)} \geq p_j^{(i-1)} \ \forall j \in [n]$ to obtain optimal $p^{(i)}$.
 4:     Post-process $p^{(i)}$ with Algorithm 3.
 5: **return** $p^{(k)}$

---

**Proposition D.1** (Monotonicity in budget $k$). *Algorithm 7 guarantees that $p^{(i)} \geq p^{(i-1)}$ for all $i \in [k]$.*

*Proof.* Let $p^{(i-1)}$ be the output at step $i-1$ and $p'^{(i)}$ be the output for iteration $i$ before post-processing and $p^{(i)}$ be the final output after post-processing. If an interval $b$ is strictly below a set of intervals $A$, either $p_b^{(i-1)} = 0$ or $p_b^{(i-1)} > 0$ and $p_a^{(i-1)} > 0 \ \forall a \in A$. If $p_b^{(i-1)} = 0$, clearly $p_b^{(i)} >= p_b^{(i-1)}$. If $p_b^{(i-1)} > 0$ and $p_a^{(i-1)} > 0 \ \forall a \in A$, then $p_a'^{(i)} = 1$ for all $a \in A$ and $p_b'^{(i)} \geq p_b^{(i-1)}$, so the post-processing will not decrease $p_b'^{(i)}$ at all and monotony will be satisfied. $\square$

## E  Existing Randomized Selection Algorithms

In this section, we provide precise pseudocode for existing algorithms against which we compare our MERIT approach in Section 5. Note that for Randomize Above Threshold, $T$ may be chosen independently of the data (e.g., as an absolute quality cutoff) or based on the data (e.g., as the $k$-th largest lower bound).

---

**Algorithm 8** Swiss NSF Selection Algorithm [21]

---

**Require:** Set of proposals with point estimates and intervals; number accepted $k$.

 1: Rank proposals by decreasing point estimate and let $e_{(k)}$ be the point estimate of the $k$-th ranked proposal.
 2: Let $\mathcal{A}$ be the set of proposals with lower bound strictly above $e_{(k)}$, $\mathcal{R}$ be the set of proposals with upper bound strictly below $e_{(k)}$, and $\mathcal{P}$ the set of proposals with intervals that contain $e_{(k)}$.
 3: Accept $\mathcal{A}$, reject $\mathcal{R}$, and accept $(k - |\mathcal{A}|)$ proposals chosen uniformly at random from $\mathcal{P}$.

---

**Algorithm 9** Randomize Above Threshold

---

**Require:** Set of proposals with point estimates and intervals; number accepted $k$.

1: Choose a threshold $T$ (potentially based on the data.)
2: Reject all intervals strictly below $T$ and select uniformly at random among the remaining.

---

## F   Details of Interval Generation

In this section, we provide additional details on how intervals may be generated.

### F.1   Interval Generation Methods

Below, we describe five potential methods of producing "Knightian" uncertainty intervals, where a funder does not draw probabilistic conclusions about the intervals.

1. *Imputation-based intervals:* in many grant funding panels most proposals have been reviewed by a large fraction of the reviewers. For example, in the Swiss NSF process [21], over $91\%$ pairs of reviewer-proposal pairs received scores. If every reviewer scored every proposal, then the funder need not worry about reviewer miscalibration—the tendency for different reviewers to interpret the review scale differently[59]. Hence, the funder may first impute values for missing scores and then aggregate across reviewers. The funder can make minimal assumptions by adopting Manski bounds [34, 47]. Manski bounds generate *intervals* by imputing missing scores with a range of possible values from the minimum to maximum score. The aggregate scores are then given as an interval over the range of possible imputed values. Hence, the intervals represent the plausible set of values for each proposal's aggregate review score, without a probabilistic interpretation.

2. *Intervals from model ensembling:* in many cases, a funder may have a number of plausible models of their data, each of which produces point estimates. Hence, the funder can estimate all plausible models of the data and ensemble into intervals by taking quantiles of the point estimates. This type of ensembling has been applied to time-series forecasting problems where it is known as Quantile Prediction Averaging [39].

3. *Intervals based on expert input:* a frequently cited motivation for randomization is concerns about prejudice, for example, against highly original ideas or junior researchers. The funder may not be able to reliably estimate such sources of error from their observational, potentially sparse data, but can rely on prior controlled experiments that establish the rough magnitude of prejudices in the review process.

4. *Multi-criteria aggregation:* Reviewers are often given a number of criteria on which proposals are rated, for example "intellectual merit" and "broader impact." The funder can consider multiple valid ways to aggregate criteria and generate intervals over the set of possible aggregations.

5. *Robustness to mis-specification of model used to generate intervals:* the funder may use a probabilistic model to estimate intervals and then draw conclusions about the ordering of the intervals, but may not trust other information about the distribution. This is a common assumption in the literature on distributionally robust optimization (see [41] for a survey), where the funder is said to be optimizing over a "support-only" ambiguity set.

### F.2   Implementation of Interval Generation in Experiments

Our worst-case objective (1), assumes that any ordering consistent with quality intervals could be the true ranking of proposals. The utility is then defined as the worst-case expected fraction of top-$k$ proposals chosen over all possible orderings consistent with the intervals. We simulate generating such quality intervals in three scientific peer review scenarios and measure the performance of different selection methods with respect to our worst-case objective. First, we replicate the scientific grant funding process of the Swiss NSF. We use publicly released review data from their 2020 grant review process consisting of $n = 353$ proposals [21]. We generate intervals using their linear model of reviewer miscalibration. Additionally, we generate imputation-based intervals for the Swiss NSF data using Manski bounds to impute missing reviewer-proposal scores and aggregating scores across reviewers using the median. Second, we use paper review data from the NeurIPS 2024 conference,

available on OpenReview, which includes $n = 4035$ accepted papers. We simulate a process of allocating long talks (orals and spotlights) among accepted papers, inspired by the USENIX Security 2025 [57]. Third, we use paper review data from the ICLR 2025 conference, available on OpenReview, which includes all $n = 11520$ papers submitted to the conference. We simulate allocating paper acceptances at this conference. For NeurIPS and ICLR, there is no standard method to generate intervals. We therefore implement three different methods for generating intervals on NeurIPS and ICLR data:

(1) *Leave-one-out intervals (LOO)*: taking inspiration from "jacknife" or leave-one-out intervals [6], we compute the range of possible mean review scores, leaving out one reviewer at a time for each paper.

(2) *Gaussian error model credibility intervals*: similar to the Swiss NSF's intervals, we assume a Gaussian model generates review scores based on underlying true quality scores and infer credibility intervals for the true quality. Specifically, for paper $i$, we assume paper has true quality $\theta_i \sim \mathcal{N}(0, 2)$, precision $\tau_i \sim \text{Gamma}(1, 1)$, and review scores on the paper are drawn i.i.d. from $\mathcal{N}(\theta_i, 1/\sqrt{\tau_i})$. The parameters of the priors are chosen to closely match those of the Swiss NSF model. We infer $50\%$ credibility intervals for $\theta_i$ given the observed review scores using MCMC.

(3) *Subjectivity intervals*: NeurIPS 2024 and ICLR 2025 both asked reviewers to provide numerical scores of papers' soundness, presentation, and contribution in addition to overall scores. Previous works have observed that different reviewers may have different subjective views of which criteria matter to a paper's quality introducing arbitrariness into overall review scores [28]. One proposed approach to mitigate this subjectivity is to learn a mapping from sub-criteria to an overall score based on peer review data [38] and then use this mapping to adjust the review score. We generate intervals by applying this method to adjust scores and taking the interval to be all values in between original scores and subjectivity adjusted scores.

We note that there are no confidence intervals for estimates generated in the worst-case setting as these are generated for a single dataset with a single set of intervals, where there is no randomness in the data generation process.

## G   Additional Experiments

In this section, we provide additional experimental results comparing the performance of randomized selection rules.

### G.1   Ablations

We additionally conduct ablations on parameters of both models that provide insight into the relative performance of different methods under varying model settings. First, in Figure 4, we show utility of different methods in the Swiss NSF's linear miscalibration model varying the degree of miscalibration. As the magnitude of miscalibration increases, deterministic selection using mean score degrades greatly in performance it does not account for error due to miscalibration at all. Meanwhile, the other methods perform similarly and maintain fairly high expected utility, even with a large degree of miscalibration. We observe similar results in additional ablations of the miscalibration model and when selecting the top one-tenth of proposals instead of top one-third, as shown in Appendix G.

In Figure 5, we test Manski bounds on the Swiss NSF dataset of grant reviews, where we impute missing values with the full range of scores. We artificially drop reviews to increase the sparsity of review scores, leading to worse utility. With 40% sparsity, deterministic achieves near zero utility as almost all intervals overlap. At all sparsity levels, MERIT outperforms both Swiss NSF and Deterministic selection. In fact, at sparsity of 0.15 to 0.25, the optimality gap between Swiss NSF and MERIT is the same as that of deterministic selection and MERIT.

### G.2   Qualitative Comparison of Outcomes

In order to give insight into how differing approaches may lead to different types of lotteries in actual peer review settings, we present qualitative differences between MERIT and the Swiss NSF approach.

In Table 1, we provide a high-level comparison of the marginal probabilities of the sampling proposals given under MERIT and the Swiss NSF algorithm. One simple comparison point is on the number of

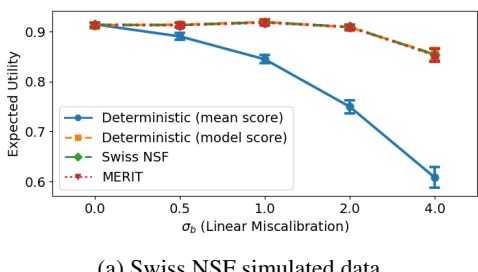

(a) Swiss NSF simulated data.

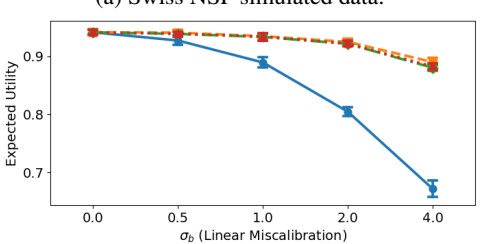

(b) Conference simulated data.

Figure 4: Ablation study comparing methods under the Swiss NSF's model of linear miscalibration with varying levels of miscalibration. Error bars show bootstrapped 95% CIs for the sample mean over 50 samples of randomly generated data from the model.

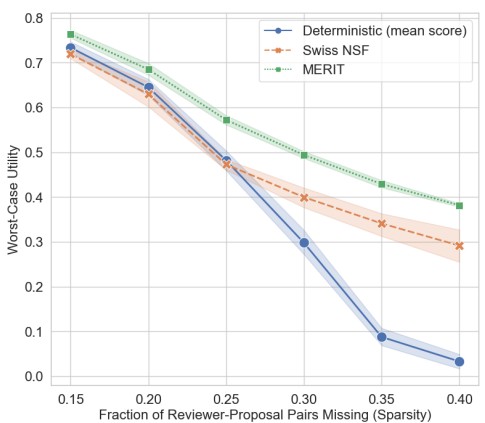

Figure 5: Worst-case utility over Manski bound intervals as a function of the fraction of reviewer-proposal pairs missing, with random dropping of review scores to increase sparsity for the Swiss NSF panel review dataset. Error bars show bootstrapped 95% CIs for the sample mean over 50 trials of randomly dropping review scores.

proposals that entered into a lottery (% Random) under each algorithm. We find that MERIT tends to randomize over an equal or greater number of candidates than the Swiss NSF algorithm—on Swiss NSF data, NeurIPS LOO, NuerIPS Gaussian, and ICLR Subjectivity the two algorithms randomize over a similar number of candidates, while on NeurIPS Subjectivity, ICLR LOO, and ICLR Gaussian, MERIT randomizes over more candidates than Swiss NSF. The biggest difference between the two approaches with respect to the outcome, is that the Swiss NSF rule assigns the same probability to every proposal that enters its lottery tier, whereas MERIT allows a range of probabilities: the broadest span is on NeurIPS Gaussian where marginal probabilities range from 0.16 to 0.83. This reflects the additional flexibility of MERITś lottery. Finally, the relative weight MERIT puts in the "certain accept" group compared to Swiss NSF ($p = 1$) is not uniform—it is higher than Swiss NSF for NeurIPS Subjectivity, essentially tied on the original Swiss NSF set, and lower for the two ICLR variants, so the split between guaranteed and lottery funding depends on the structure of the intervals in each study rather than following a single trend.

| Dataset | MERIT | | | Swiss NSF Algorithm | | |
|---|---|---|---|---|---|---|
| | % Accept | % Random | Range of $p$ | % Accept | % Random | Value of $p$ |
| Swiss NSF | 28.3 | 3.4 | (0.5, 0.5) | 28.0 | 3.7 | 0.54 |
| Swiss NSF (Manski Bounds) | 5.4 | 26.9 | (0.75, 0.94) | 5.4 | 26.9 | 0.92 |
| NeurIPS LOO | 3.4 | 16.5 | (0.36, 0.94) | 3.4 | 17.9 | 0.34 |
| NeurIPS Gaussian | 2.2 | 25.7 | (0.16, 0.83) | 4.0 | 27.5 | 0.20 |
| NeurIPS Subjectivity | 4.5 | 18.7 | (0.14, 0.45) | 1.6 | 10.4 | 0.78 |
| ICLR LOO | 11.1 | 32.4 | (0.51, 0.88) | 22.0 | 22.8 | 0.44 |
| ICLR Gaussian | 9.5 | 34.4 | (0.45, 0.87) | 21.2 | 27.7 | 0.4 |
| ICLR Subjectivity | 17.6 | 24.3 | (0.50, 0.88) | 16.2 | 25.2 | 0.63 |

Table 1: Comparison of marginal probabilities of acceptance by MERIT and Swiss NSF on each dataset. "Accept" = guaranteed to be selected ($p = 1$), while "Random" = entered into lottery ($0 < p < 1$).

# H  Additional Analysis of Computational Runtime Case Studies

In Figure 6, we show the runtime in seconds of the MERIT algorithm (including all pre-processing and post-processing steps) on each dataset as a function of the acceptance rate. For all methods, the algorithm runs in under five minutes. Runtime increases with acceptance rate, which is expected because the number of constraints grows with the number of selections $k$. We find that the cutting plane algorithm converges in under 30 iterations for all datasets, meaning that it solves under 30 linear programs. Furthermore, the largest linear program solved has 25,000 constraints when choosing $k = 5760$ of the $n = 11520$ ICLR papers, suggesting that the cutting plane algorithm scales well in $n$ and $k$.

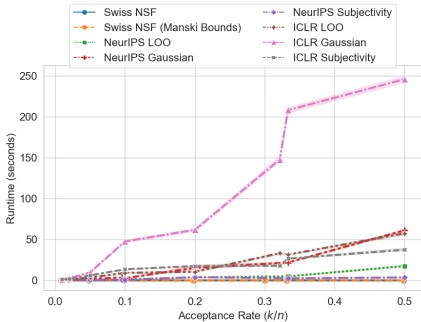

Figure 6: Runtime of MERIT on a standard personal laptop (in seconds) as a function of acceptance rate using review data from the Swiss NSF ($n = 353$), NeurIPS 2024 accepted papers ($n = 4035$) and ICLR 2025 papers ($n = 11520$).

In Figure 7, we present the number of cuts and the number of iterations it takes for the cutting plane algorithm to converge. We note that the size of the LP solved and convergence rate could potentially be optimized further by strategically pruning cuts from the linear program at each iteration, but even without additional optimizations the algorithm yields practical performance.

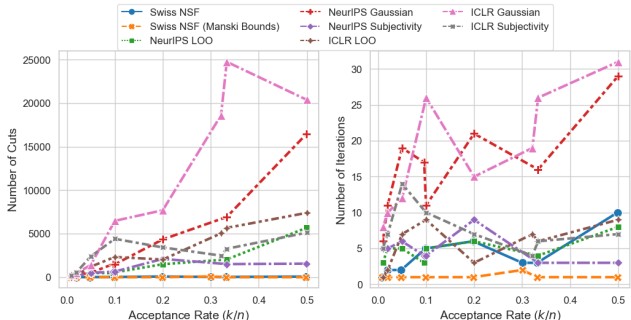

Figure 7: Convergence of cutting plane algorithm as a function of acceptance rate using review data from the Swiss NSF ($n = 353$), NeurIPS 2024 accepted papers ($n = 4035$) and ICLR 2025 papers ($n = 11520$). The number of cuts corresponds to the size of the largest linear program solved in a single iteration of the cutting plane algorithm and the number of iterations corresponds to the total number of LPs solved before convergence.

# I  Ex-Ante Optimal Uniform Random Lottery

An additional benefit of MERIT is that it can be easily adapted to handle constraints on the form of the lottery. In some contexts, funders or decision-makers may prefer to implement a uniform lottery, where all candidates subject to randomization are selected with equal probability. This form of randomization may be viewed as simpler, more transparent, and more acceptable to participants than one in which probabilities differ across candidates.

To accommodate such requirements, the optimization problem underlying MERIT can be modified to constrain all randomized candidates to share a common probability of selection. This yields the *ex ante optimal uniform lottery that still satisfies the ex post validity constraints.* The resulting optimization problem is given below:

$$\max_{p \in \mathbb{R}^n, v \in \mathbb{R}, c \in \mathbb{R}} v$$

$$\text{subject to} \quad v \leq \sum_{i=1}^{n} p_i \mathbb{1}\{\sigma(i) \leq k\}, \quad \forall \sigma \in \Sigma_n, \tag{4}$$

$$\sum_{i=1}^{n} p_i = k \text{ and } 0 \leq p_i \leq 1, \forall i \in [n]$$

$$p_i \in \{0, 1, c\}, \quad \forall i \in [k] \tag{5}$$

$$p_i = 1 \text{ or } p_j = 0 \quad \forall i, j \in [n] : \ell_i > u_j \tag{6}$$

The optimization problem is equivalent to the original optimization problem 2, with the addition of constraint (5), which forces the lottery to be uniform, and constraint (6), which ensures ex post validity. These additional constraints turn the linear program into a mixed integer program (MIP). Notably, this MIP has polynomial in $n$ constraints, excepting the worst-case ordering constraints (4) present in the original optimization problem. Hence, this can be solved using Algorithm 2 (the MERIT cutting plane algorithm) initialized with the additional integer constraints.

**Empirical Performance.** Figure 8 compares the uniform variant of MERIT (**MERIT Uniform**) against the base MERIT and the Swiss NSF mechanism across datasets. The results show that the uniform constraint has minimal effect on expected utility, while preserving or improving worst-case robustness under our model.

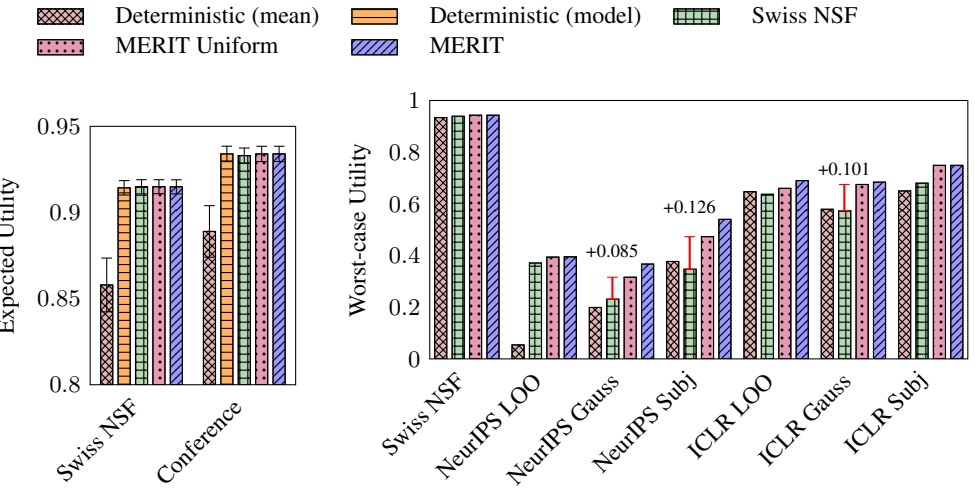

(a) Linear miscalibration model.  (b) Worst-case over interval ordering model (our model).

Figure 8: Proportion of top-$k$ proposals selected by different methods with quality data generated under the Swiss NSF model of linear miscalibration and under our model of worst-case over feasible rankings. MERIT Uniform matches performance of algorithms designed for the Swiss NSF's linear model and recovers much of the gap between MERIT and other methods in the worst-case over intervals defined by our model

Table 2 compares the qualitative outcomes under different algorithms. Notably, MERIT Uniform achieves nearly identical marginal acceptance probabilities and competitive runtime performance, suggesting that funders can adopt this more interpretable form of randomization without substantial efficiency loss.

| Dataset | MERIT | | | Swiss NSF | | | MERIT Uniform | | |
|---|---|---|---|---|---|---|---|---|---|
| | % Acc | % Rand | $p$ | % Acc | % Rand | $p$ | % Acc | % Rand | $p$ |
| Swiss NSF | 28.3 | 3.4 | 0.5–0.5 | 28.0 | 3.7 | 0.54 | 28.3 | 3.4 | 0.5 |
| NeurIPS LOO | 3.4 | 16.5 | 0.36–0.94 | 3.4 | 17.9 | 0.34 | 3.8 | 16.1 | 0.36 |
| NeurIPS Gaussian | 2.2 | 25.7 | 0.16–0.83 | 4.0 | 27.5 | 0.20 | 4.9 | 19.3 | 0.24 |
| NeurIPS Subjectivity | 4.5 | 18.7 | 0.14–0.45 | 1.6 | 10.4 | 0.78 | 3.6 | 16.1 | 0.37 |
| ICLR LOO | 11.1 | 32.4 | 0.51–0.88 | 22.0 | 22.8 | 0.44 | 11.1 | 25.2 | 0.83 |
| ICLR Gaussian | 9.5 | 34.4 | 0.45–0.87 | 21.2 | 27.7 | 0.40 | 17.9 | 25.6 | 0.56 |
| ICLR Subjectivity | 17.6 | 24.3 | 0.49–0.88 | 16.2 | 25.2 | 0.63 | 23.0 | 18.4 | 0.49 |

Table 2: Comparison of marginal probabilities of acceptance by MERIT, Swiss NSF, and MERIT Uniform on each dataset. "Acc" = guaranteed to be selected ($p = 1$), while "Rand" = entered into lottery ($0 < p < 1$). MERIT shows the range of probabilities, while Swiss NSF and MERIT Uniform assign single uniform probabilities.

Finally, Table 3 compares the runtime of the original and uniform version of MERIT. The runtime of MERIT Uniform can be significantly slower than MERIT due to solving an integer program, but still runs in reasonable time, running in 40 mins instead of 4 mins in the slowest case of ICLR Gaussian, which has over 10,000 candidates from which to choose. Additionally, we performed no additional optimizations of solver parameters in our evaluations of MERIT Uniform, so there may be further speed ups that are possible.

Table 3: Runtime Comparison for MERIT and MERIT Uniform

| Dataset | Runtime (seconds) | |
|---|---|---|
| | MERIT | MERIT Uniform |
| Swiss NSF | 0.046 | 0.043 |
| Swiss NSF (Manski Bounds) | 0.029 | 0.036 |
| NeurIPS LOO | 0.941 | 1.301 |
| NeurIPS Gaussian | 5.618 | 43.278 |
| NeurIPS Subjectivity | 1.883 | 11.319 |
| ICLR LOO | 43.561 | 194.366 |
| ICLR Gaussian | 242.222 | 2,346.693 |
| ICLR Subjectivity | 21.060 | 22.225 |

