# OpenReview forum: "A Principled Approach to Randomized Selection under Uncertainty: Applications to Peer Review and Grant Funding"
_NeurIPS.cc/2025/Conference — NeurIPS 2025 spotlight_

### Official Review · Reviewer_CFrv · 2025-06-24

**Clarity:** 3
**Significance:** 3
**Originality:** 3
**Rating:** 5
**Confidence:** 3

**Summary:**

The paper is motivated by the need to select top choices from a large group of candidates, where the quality against some qualifying criteria is uncertain. Specifically, this issue is present in peer-review journals and funding proposals, where assessors can be uncertain about the quality of a submission. This paper proposes a polynomial time algorithm which, given estimated quality intervals, returns a quantity of top-proposals whilst maintaining some key assumptions. The paper proves these desired properties, compares with the current literature, and demonstrates practical applicability with peer review data.

**Questions:**

Can you give an intuition as to why the acceptance rate between the two algortihms is more varied for the ICLR dataset? (Table 1).

Does the edge-case of k=1 still maintain the stated properties? Specifically, its seems to me that the maximum instability for MERIT could occur in this instance, but would be keen to hear your thoughts.

**Ethical Concerns:**

["NO or VERY MINOR ethics concerns only"]

**Final Justification:**

No unresolved issues after discussion, I confirm my recommendation.

**Limitations:**

Yes

**Quality:**

3

**Strengths And Weaknesses:**

The paper is well-motivated with the limitations of related work explained, and the benefits of this apporach are clearly presented. There is a good level of discussion in the comparison between algorithms, which is grounded in both theoretical and empirical analysis.

In Section 5, the discussion on violating the monotonicity in budget could be further developed, perhaps an intuition as to why the current algorithms do not hold this property. Appendix A, Figure 2 would benefit from axis labels to make it more understandable.

---

> ### Author Rebuttal · Authors · 2025-07-31
>
> Thank you for your feedback and interesting questions about our axiomatic comparison. Below we address some of your questions:
>
> * Figure 2 in Appendix A gives the main intuition for why MERIT violates monotonicity. We will add clearer axis labels. The Figure has four intervals where interval 1 lies entirely above intervals 3 and 4, while intervals 2,3, and 4 overlap.  When k=1, the top 1 proposal (given the overlaps in the intervals) could either be proposal 1 or proposal 2 since these two intervals overlap, while 3 and 4 are strictly below 1. Hence, MERIT chooses uniformly at random between proposals 1 and 2. Meanwhile, when k=2, proposal 1 is always in the top 2 so it gets accepted with probability 1. This leaves 1 probability mass to allocate between proposals 2,3,4. Since any of these 3 proposals could be the other proposal in the top-2, they each receive probability 1/3 of acceptance. Therefore, proposal 2 is worse-off when selecting 2 proposals rather than 1 proposal.
> Can you give an intuition as to why the acceptance rate between the two algortihms is more varied for the ICLR dataset? (Table 1).
>
> * In our formal definition of “maximal instability” we specifically define the axiom for budget (k) ranging from 2 to n-2, because k=1 is not a particularly meaningful case to consider for the stability of selection algorithms. Just by definition when k=1, changing one proposal (highest one) can fully change the outcomes. This holds even choosing deterministically or for any other reasonable method. Hence k>1 is a more meaningful parameter range in which to consider the stability of an algorithm. We will clarify this in the main text.

---

> > ### Comment · Reviewer_CFrv · 2025-08-06
> >
> > Thank you for your responses. The suggested clarifications are needed to make the paper accessible so I advise you to implement them.

---

### Official Review · Reviewer_Api3 · 2025-06-30

**Clarity:** 3
**Significance:** 3
**Originality:** 3
**Rating:** 4
**Confidence:** 4

**Summary:**

This paper introduces MERIT, which is a randomized decision-making framework for evaluating and selecting items under uncertainty. The core idea is to use interval estimates of item quality rather than precise point estimates to model the inherent uncertainty in domains like peer review, hiring, and admissions. The authors propose a maximin optimization problem where the goal is to maximize the worst-case expected number of top candidates selected, considering all possible true rankings consistent with the given quality intervals (Knightian uncertainty). They develop a polynomial-time algorithm for MERIT, and demonstrate it scales reasonably on real-world datasets. Additionally, they provide an axiomatic comparison with existing ad hoc randomization methods and show how to ensure ex-post validity through a post-processing step.

**Questions:**

See weakness

**Ethical Concerns:**

["NO or VERY MINOR ethics concerns only"]

**Final Justification:**

I did not give it a score of 5 because of my concern that the axiomatic criteria are based on a worst-case scenario, which is unlikely to happen in practice. I would have appreciated more practically relevant axiomatic criteria, which I believe can be studied under this model.

**Limitations:**

Yes,

**Quality:**

3

**Strengths And Weaknesses:**

Strengths
* The paper proposes a optimization-based framework for randomized selection, which seems much better than ad-hoc methods, and incorporates uncertainty through interval estimates and does not make strong model-based assumptions.
* The maximin objective function, maximizing the worst-case expected number of top-k candidates, is interesting.
* The paper demonstrates that the proposed algorithm is computationally practical.
* Axiomatic Analysis against properties like "monotonicity in budget" and "stability" is interesting
 * Explicitly adding a post-processing step to guarantee ex-post validity (ensuring a superior item is selected if an inferior one is) seems like a reasonable thing to add to the proposed mechanism.

Weaknesses:
* I am not an expert in this area, but the core maximin LP formulation appears to be a re-invention of well known min-max or Stackelberg problems, which have been extensively studied in various related settings. The paper does not clearly position its maxmin formulation against existing works on Stackelberg/min‑max settings.

* I'm a bit confused by lines (112-114): "For example, in the Swiss NSF ... 91% pairs of reviewer-proposal pairs received scores .... reviewer miscalibration."
Does "over 91% pairs of reviewer-proposal pairs received scores" actually mean that, on average, each reviewer reviewed almost every proposal (like, say, 910 out of 1000 proposals)? Assuming it's not true, the usage for Manski-bound intervals seems rarely achievable in typical peer-review environments. Is this the they were not used for ICLR/Neurips experiments?

* Axiomatic Criteria: One of the proposed axioms "maximal instability" (a tiny interval tweak causing a flip from fully deterministic to fully random) also appears to be a contrived, pathological case that may not reflect typical real-world intervals (as the example considers only three possible values of range for n proposals). More practically relevant axiomatic criteria like robustness against a range of adversarial behaviors like axioms limiting the opportunity for strategic manipulation by a single or subset of reviewers, would have been better.


* Post-Processing: Ex-post validity processing could, in principle, be applied to other baseline mechanisms. Is MERIT intrinsically better than other mechanisms (in some way)?

* Underutilization of Interval information: (Not a weakness in a real sense) While the algorithm is designed for Knightian uncertainty, the maximin approach seems to underutilize information within interval overlaps. For example, given intervals like [1,5] and [4,8], a human might intuitively feel B is "more likely" to be better,

---

> ### Author Rebuttal · Authors · 2025-07-31
>
> Thank you for the review and for your attention to the paper's comparison to existing methods. We are glad that you found the problem formulation and axiomatic analysis interesting. Below we address some of your concerns:
>
> 1. **Novelty of the formulation:** While Stackelberg / min-max formulations are indeed well-established and widely used in many different applications, our paper makes two novel contributions:
>    - First, as we discuss in Section 4 of the paper, it is computationally challenging to solve for an equilibrium in the specific Stackelberg game we describe. Many related optimization problems—minimum weight k-ideal problem [FK94], maximum cut on interval graphs [ABMR23], interval graph completion [RAK91] and minimum weighted completion time of jobs subject to interval ordering precedence constraints [Woe03]—are known to be NP hard. Our work develops a novel, polynomial-time algorithm to solve for the Stackelberg equilibrium in our problem setting and this algorithm is also practically efficient.
>    - Second, to our knowledge, modeling uncertainty in selection decisions as a Stackelberg game has not been done previously and represents a new take on the problem.
>
> 2. **Manski bounds:** The statement in the paper about the use of Manski bounds on the Swiss NSF data is indeed accurate. The Swiss NSF setting only has ~300 proposals and panels meet and discuss each proposal and then vote on all proposals. Hence, the only reason for missing evaluations is conflict-of-interest. Therefore, the Manski intervals indeed cover most review pairs.
>
>    For conference settings, like NeurIPS and ICLR you are correct that Manski bounds would result in meaningless intervals that cover the entire range of scores since there are few reviews per paper, so we do not use Manski bounds there. We test a number of methods for generating intervals in conference settings, detailed in Appendix F.
>
> 3. **Axiomatic criteria:** Our interpretation of the maximum instability axiom is that all natural algorithms should obey even this very weak notion of stability. It was quite surprising to us that existing methods did not respect this axiom and could be "maximally unstable."
>
> 4. **Post-processing for monotonicity:** We agree that other algorithms could also be post-processed in order to enforce monotonicity. MERIT is intrinsically better with respect to the robust objective over interval uncertainty and also is guaranteed to meet the ex-post validity principle, which prior methods do not guarantee. Additionally, our algorithm to enforce monotonicity for MERIT empirically seems like it does not lead to much worse performance with respect to worst-case ex ante optimality. In the modified algorithm, we obtain a locally optimal solution at each step (i.e., at budget k we solve for the ex ante optimal vector of probabilities that is monotonic with respect to the solution obtained at budget k-1). It will be an interesting area for future work to try to prove that this algorithm (or a suitable modification) is near the global optimum when one wishes to maximize the ex ante objective with the additional constraint of monotonicity.
>
> ## References
>
> [FK94] Ulrich Faigle and Walter Kern. Computational complexity of some maximum average weight problems with precedence constraints. Operations Research, 42(4):688–693, 1994.
>
> [ABMR23] Ranendu Adhikary, Kaustav Bose, Satwik Mukherjee, and Bodhayan Roy. Complexity of maximum cut on interval graphs. Discrete & Computational Geometry, 70(2):307–322, 2023.
>
> [RAK91] Ramamurthy Ravi, Ajit Agrawal, and Philip Klein. Ordering problems approximated: Single-processor scheduling and interval graph completion. In Automata, Languages and Programming: 18th International Colloquium Madrid, Spain, July 8–12, 1991 Proceedings 18, pages 751–762. Springer, 1991.
>
> [Woe03] Gerhard J Woeginger. On the approximability of average completion time scheduling under precedence constraints. Discrete Applied Mathematics, 131(1):237–252, 2003.

---

> > ### Comment · Reviewer_Api3 · 2025-08-09
> >
> > Thank you for the rebuttal and the additional references. I have now increased my score

---

### Official Review · Reviewer_iwWb · 2025-07-02

**Clarity:** 3
**Significance:** 4
**Originality:** 3
**Rating:** 6
**Confidence:** 5

**Summary:**

The paper formalises the problem of *top-\\( k \\)* selection under Knightian uncertainty by modelling each candidate with an estimated quality interval and requiring the decision-maker to maximise the *worst-case* expected number of true top-\\( k \\) items selected.
This leads to a max--min linear program whose feasible set is defined by the partial order induced by the intervals.
The formulation explicitly encodes two criteria: *ex-ante optimality* (robust utility maximisation) and *ex-post validity* (no dominated candidate may be selected).

To solve the intractable LP, the authors develop MERIT (Maximin Efficient Randomised Interval Top-\\( k \\)), which combines:

- a polynomial-time separation oracle tailored to interval orders, and
- a practical cutting-plane implementation that converges in under 30 LP solves, even for \\( n > 10{,}000 \\) proposals.

The result is the first algorithm with provable polynomial complexity for this class of robust selection problems (Theorem 4.1).

The paper introduces two behavioural axioms: *monotonicity in budget* and *stability*, and proves that MERIT avoids "maximal instability," a pathology observed in existing peer-review lotteries, while performing comparably on monotonicity (which is violated by all known randomised methods).

Experiments on Swiss NSF grants and NeurIPS/ICLR conference data show that MERIT scales to real workloads and achieves higher worst-case utility than either deterministic top-\\( k \\) or the Swiss NSF rule, while enabling a more nuanced allocation of lottery probabilities.

**Questions:**

- The construction of the quality intervals could be elaborated further.
  In particular, I wonder whether the constructed intervals tend to be large, and whether the true quality distribution within the interval may be skewed, especially toward the lower end.

**Ethical Concerns:**

["NO or VERY MINOR ethics concerns only"]

**Final Justification:**

Dear AC and authors,

I really like the paper. I would recommend this paper for an oral presentation, given the potential impact it could have across a range of applications.

Best wishes,
Your reviewer

**Limitations:**

- MERIT is substantially more complex to communicate than a simple uniform lottery.
  Stakeholders may be reluctant to adopt a mechanism that requires explaining a max--min LP, cutting-plane optimisation, and post-processing steps.

**Quality:**

4

**Strengths And Weaknesses:**

I found the paper very well-written and compelling.
It presents a powerful optimisation framework with broad applicability to important real-world problems.
The results are theoretically sound and empirically impressive.

**weakness**

- MERIT is substantially more complex to communicate than a simple uniform lottery.
  Stakeholders may be reluctant to adopt a mechanism that requires explaining a max--min LP, cutting-plane optimisation, and post-processing steps.

- The construction of the quality intervals could be elaborated further.
  In particular, I wonder whether the constructed intervals tend to be large, and whether the true quality distribution within the interval may be skewed, especially toward the lower end.

---

> ### Author Rebuttal · Authors · 2025-07-31
>
> Thank you for the thoughtful and detailed review, and for recognizing the theoretical and practical contributions of the work.
>
> 1. **Constructing intervals:** Our approach is designed to not be too closely tied to any specific model, so there are a number of ways that a decision maker could generate intervals. We had limited space to describe approaches for interval generation in the main text but give more detail in Appendix F on methods for generating intervals in our experiments. For example, the funder could simply take intervals as the range from minimum and maximum reviewer score given to each proposal or they could compute bootstrapped estimates. Alternatively, they could accommodate subjectivity in the reviewer recommendations by allowing for different mappings from sub-criteria scores ("quality", "clarity", "significance", and "originality" in NeurIPS reviews) to an overall score.
>
> 2. **Interval characteristics:** The width of the constructed intervals varies between methods and settings. Below, we provide the mean width of intervals for the methods we tested (after normalizing the units of all intervals to lie in [0,1]):
>
>     | Dataset | Average Width |
>     |---------|---------------|
>     | Swiss NSF | 0.0625 |
>     | Swiss NSF (Manski Bounds) | 0.0198 |
>     | NeurIPS LOO | 0.1259 |
>     | NeurIPS Gaussian | 0.1671 |
>     | NeurIPS Subjectivity | 0.0700 |
>     | ICLR LOO | 0.0917 |
>     | ICLR Gaussian | 0.1196 |
>     | ICLR Subjectivity | 0.0684 |
>
>     In terms of the skew of the intervals, because we do not know the ground truth for the "true quality" of each candidate in real-world datasets like ICLR and NeurIPS, it is not possible to measure where in the interval the true quality lies. This is the motivation for optimizing over intervals where the true quality could lie anywhere within each interval.
>
> 3. **Complexity of communicating MERIT:** This concern is valid, as we discuss in the Limitations section, but we have reasons to be optimistic about adoption of MERIT. First, funders are using the current Swiss NSF method—this method has a simpler decision making part than MERIT, but a much more complex method of generating intervals (hence the overall method is complex.) In spite of the complexity of the Swiss NSF's interval generation method it has been put into use for real-world funding decisions. Second, we are currently speaking with organizers of such selection processes, and so far, complexity has not come up as an issue.

---

### Official Review · Reviewer_XMKU · 2025-07-05

**Clarity:** 4
**Significance:** 3
**Originality:** 4
**Rating:** 5
**Confidence:** 3

**Summary:**

The scope of this work is the design of a novel randomized selection mechanism that improves upon previous approaches under multiple aspects.
More precisely, the authors consider settings where a funder receives $n$ proposals and has to estimate the quality of each one based on the available data via intervals $[\\ell\_i, u\_i]$ for $i \\in [n]$.
The intervals essentially represent the inherent uncertainty of the funder, who then has to select $k \\le n$ of the received proposals.
After reviewing some of the mainly adopted selection criteria and pointing out their shortcomings, such as the lack of maximal instability, the authors introduce a novel randomized selection algorithm called MERIT, whose distribution over proposals is computed by solving a linear program with combinatorially many constraints.
The authors then illustrate that it is possible to efficiently compute solutions to the same optimization problem in polynomial time, and provide experiments that confirm their theoretical findings.
Finally, the work provides general results about randomized selection algorithms via an axiomatic approach that is reminiscent of results from social choice theory, as motivated by the authors themselves.

**Questions:**

- Could you expand a bit on the relationship with related work that is briefly discussed in the paragraph “Selection Under Uncertainty and Robust Optimization” at lines 332-340? In particular, could you provide more details as to why those techniques could not be adapted to the problem setting considered in this work?
- The results in Section 5, and more precisely point (3) of Theorem 5.1, demonstrate the impossibility of guaranteeing both the ex-ante optimality and the monotonicity in budget properties. Do you think this impossibility result can be avoided by considering slight relaxations of the two axioms, namely approximate ex-ante optimality (given an additive/multiplicative approximation factor, or some other meaningful relaxation of the optimization problem) or $\\epsilon$-approximate monotonicity in budget (i.e., increasing the number of proposals cannot decrease the selection probability by more than $\\epsilon$, in place of just strict monotonicity)? What about other reasonable but slightly weaker axioms?
- Theorem 4.3 is stated to hold only for ex-ante optimal solutions $p$. Does the statement actually hold for any solution $p$? At least, from the discussion below, it seems that the claim could hold for any feasible (and not necessarily optimal) ex-ante solution $p$, thus stating that the ex-ante objective does not worsen instead of simply stating that ex-ante optimality is preserved if it already holds. If so, then the result of Theorem 4.3 could be stated with more generality.

**Ethical Concerns:**

["NO or VERY MINOR ethics concerns only"]

**Final Justification:**

The authors fully addressed my concerns and provided replies to my questions. Since they also committed to include the required changes in their paper, I am more confident that this submission should be considered for acceptance. Hence, I increased my score accordingly.

**Limitations:**

My only comment regarding limitations is that this work would benefit from a more thorough discussion on the fairness aspects of the proposed methodology, since it could impact critical selection processes.
I believe there should be no ethical concern with respect to this work, but I would just prefer to see this point addressed in a more direct way, e.g., in the Discussion section (especially given the authors’ answer to point 2 of the Paper Checklist).

**Quality:**

3

**Strengths And Weaknesses:**

The strengths of this paper lie in its fundamentally sound and principled approach to developing a novel randomized selection mechanism. One main contribution is the detailed overview of existing state-of-the-art selection criteria like randomize-above-threshold and the Swiss NSF one. This comprehensive analysis highlights the inherent limitations of these currently adopted methods, which supports the need for a novel randomized selection algorithm as addressed in this work with the proposed MERIT algorithm. This rationale is presented with clarity, demonstrating how randomization can address the shortcomings of deterministic approaches, and more importantly, why providing a selection criterion satisfying ex-ante optimality and ex-post validity is crucial in realistic scenarios. I also particularly appreciated the axiomatic analysis in Section 5, which adds more rigor and generality to the results provided in this work. Moreover, the experimental results show how the runtime of the proposed algorithm scales well as the acceptance rate $k/n$ increases. The outline of interesting potential questions for future investigations shows that the authors have a clear understanding of the implications of their work.

Despite the significant strengths of the core methodology, some minor concerns warrant further elaboration. These concerns, as pointed out in the "Questions" below, primarily revolve around the broader implications and connections of the proposed mechanism to relevant topics. A more detailed discussion on how this new mechanism interacts with or differs from existing related notions and techniques would provide a richer context and demonstrate a deeper understanding of its place within the current research landscape. I believe addressing these points in a clearer way would strengthen the paper's overall contributions. In addition to this, I addressed some quite minor concents in the "Limitations" section that I would like to see reasonably addressed further compared to what is already present in the paper.

**Typos:**
- Line 155: should $n$ actually be $k$?
- Line 156: “game-theoretic” instead of “game theoretic”
- Line 261: missing period after “process”
- Line 265: should “maximum” actually be “maximal”?
- Line 272: “not” should be “no”
- Line 273: “thorough” should be “through”
- Line 287: missing “and” between “NeurIPS 2024” and “ICML 2025”
- Line 325: missing space between “robustness,” and “but”
- Line 1026: “ocnfirm” should be “confirm”

---

> ### Author Rebuttal · Authors · 2025-07-31
>
> Thank you for your thoughtful comments on our theoretical results and detailed attention to the writeup. Below we clarify your questions:
>
> 1. **Relation to existing work:** While our setting shares the high-level motivation of making robust decisions under uncertainty, classical robust and distributionally robust optimization techniques typically operate over convex uncertainty sets in continuous spaces (e.g., real-valued parameters that can take any value within a range). In contrast, our problem optimizes over a discrete uncertainty set of rankings: we optimize worst-case utility over a combinatorially defined set of feasible permutations consistent with interval constraints. This ranking-based uncertainty set leads to a non-convex inner problem that cannot be directly addressed by standard RO/DRO formulations.
>
> 2. **Monotonicity vs. ex-ante optimality:** This is an interesting question — we propose a simple algorithm to enforce monotonicity for MERIT in Appendix D, which allows for violation of ex-ante optimality. Empirically, it seems like it does not lead to much worse performance with respect to worst-case ex ante optimality. In the modified algorithm, we obtain a locally optimal solution at each step (i.e., at budget $k$ we solve for the ex ante optimal vector of probabilities that is monotonic with respect to the solution obtained at budget $k-1$). It will be an interesting area for future work to try to prove that this algorithm (or a suitable modification) is near the global optimum when one wishes to maximize the ex ante objective with the additional constraint of monotonicity.
>
> 3. **Generalizing Theorem 4.3:** Thanks for the great observation—yes the statement holds for any solution. Precisely, we could generalize the theorem statement as follows:
>
>    > Given a feasible solution $p$ to linear program (2), applying Algorithm 3 yields a post-processed solution $p′$ such that any sampling of $k$ proposals with marginal probabilities $p′$ satisfies ex post validity. Moreover, the objective value achieved by $p′$ is greater than or equal to that of $p$.
>
>    We intend the current statement of the Theorem to capture the usefulness of the post-processing algorithm to the MERIT algorithm, which takes as input an ex ante optimal solution to the problem. We will add the above statement as a remark to capture that the post-processing algorithm could be more generally useful.
>
> 4. **Discussion of fairness:** Our algorithm satisfies two notions of "fairness." First, one can think of the "ex-post validity" criteria as a type of fairness in randomized decisions that states that if one candidate is clearly better than another candidate, the stronger candidate should not be randomly rejected when the weaker candidate is accepted. Second, the motivation for our work starts from the premise that it is unfair that two candidates can have similar point estimates of quality, but large differences in outcomes under deterministic selection rules. Randomization is more "fair" in tie-breaking between these similar quality proposals than accepting one and rejecting another. In fact, existing literature [FC16,HTB22,GLM+25] argues that deterministic grant peer review processes introduce unfairness by tie-breaking between proposals on arbitrary, spurious signals like applicant seniority or preference for a certain subject area. We will add discussion of such fairness considerations to our Discussion section.
>
> We will make sure to fix the typos you point out in the final version of the paper.
>
> ## References
>
> [FC16] Ferric C Fang and Arturo Casadevall. Research funding: The case for a modified lottery, 2016.
>
> [HTB22] Serge PJM Horbach, Joeri K Tijdink, and Lex M Bouter. Partial lottery can make grant allocation more fair, more efficient, and more diverse. Science and Public Policy, 49(4):580–582, 2022.
>
> [GLM+25] Joanie Sims Gould, Anne M Lasinsky, Adrian Mota, Karim M Khan, and Clare L Ardern. Threats to grant peer review: A qualitative study. BMJ Open, 15(2):e091666, 2025.

---

> > ### Comment · Reviewer_XMKU · 2025-08-08
> >
> > I thank the authors for the detailed reply. This addresses my concerns and questions in a complete way.

---

### Official Review · Reviewer_BoxL · 2025-07-20

**Clarity:** 4
**Significance:** 3
**Originality:** 3
**Rating:** 5
**Confidence:** 3

**Summary:**

This paper studies the problem of selecting the top-$k$ candidates under uncertainty (e.g., in grant funding). The authors proposed a principled approach that relies on the ordering of the interval (that does not correspond to a probability measure over possible outcomes). The proposed approach, MERIT, is a polynomial-time algorithm that can approximately obtain the optimal solution.

In the absence of ground truth for evaluation, the authors propose an axiomatic approach to compare the randomized mechanisms, and demonstrate that the MERIT algorithm prevents the "maximal instability" and can achieve "monotonicity in budget" with some modifications, while existing mechanisms do not satisfy. The author further evaluated this algorithm on real-world dataset and show that the proposed algorithm can easily scales to thousands of candidates.

**Questions:**

See "Weakness" above.

**Ethical Concerns:**

["NO or VERY MINOR ethics concerns only"]

**Final Justification:**

The author response clarifies the "ex-ante optimality" principle. I don't have further questions. I keep my original rating of 5.

**Limitations:**

Yes.

**Paper Formatting Concerns:**

No.

**Quality:**

4

**Strengths And Weaknesses:**

Strengths

1. Novelty and significance. This paper studies an important problem of selecting top-$k$ candidates under noisy evaluation, which could bring great social benefits. This paper presents a novel algorithm that formalizes the idea of random selection for candidates that are statistically indistinguishable. The authors proposed ex-ante optimality and ex-post validity as two primary principles. While the induced problem is NP-hard, the authors proposed a polynomial algorithm that can obtain solution with $\epsilon$ accuracy. It is empirically verified that MERIT can be applied to large-scale, real-world problems.

2. Solid theoretical analysis and reasonable evaluation. The technical contribution is very solid, this paper formulating the selection as a maximin linear program that focuses on worst-case performance (as required by the ex-ante optimality). Solving this is technically non-trivial, and the authors derived a polynomial-time solution. Beyond the algorithm itself, the authors also proposed a way to conduct axiomatic comparison, and shown that the MERIT algorithm avoids the "maximal instability" and can be modified to satisfy "monotonicity in budget" -- for which other existing mechanisms failed to satisfy.

3. Clear presentation. I enjoyed reading this paper. While the problems and techniques are complicated, the authors presented the problem and key insights clearly. Many technical details are properly deferred to appendix.


Weakness

1. More discussion on "ex-ante optimality". This is the first principle and used to derive the LP, but I think this principle is very conservative, and needs more motivation. As described in the paper, this principle focuses on the worst-case scenario (i.e., assuming an adversary chooses a ranking after top-k is selected) -- this doesn't match with the primary motivating example of funding grant, where it's more reasonable to assume there is an unknown ground truth ranking that doesn't change after the tok-$k$ is selected (unless funding grant somehow has negative impact on research that receives the grant, which hopefully is not the case). It would be good to have more discussion on why this is a reasonable principle to have; or if there is other principle that can better approximate the "average case" instead of the "worst case".

2. Monotonicity v.s. ex-ante optimality. In theorem 5.1, the authors show that it is not possible to simultaneously satisfy ex-ante optimality and monotonicity in the budget. The monotonicity seems to be a more reasonable property, so this also makes me wondering if ex-ante optimality can be modified.

3. It's not clear to me how to compare the numbers in Table 1. It would be great if the authors can add some more discussion in the main paper on the take-away message of this table.

---

> ### Author Rebuttal · Authors · 2025-07-31
>
> Thank you for your careful reading of the paper and thoughtful comments on the problem formulation and axiomatic analysis. We address concerns below:
>
> 1. **Motivating ex-ante optimality:** this is a great point, as the average case and worst case models are the two most common modeling assumptions in the ML/CS/Statistics literature for selection and estimation problems. An “average case model” would assume that the funder has a probabilistic model of the data (e.g., a probability distribution over candidates’ true qualities and reviewers’ errors in assessing candidate quality). Our ex-ante optimality formulation starts from the assumption that a funder lacks such a probabilistic model of their data and therefore operates under Knightian uncertainty. We believe this assumption that a funder lacks a probabilistic model is appropriate for settings where quality is determined by peer review for a number of reasons motivated by prior work. First, there are many proposed probabilistic models of reviewer errors, but these models have generally performed poorly in real deployments [Sha22, Section ‘Miscalibration’]. One possible reason for this poor performance is that in practice, human miscalibration is more complex than simple models [BGK05]. Further, decision makers generally lack ground truth data with which to evaluate whether a given probabilistic model of review data is appropriate in a given setting, making it difficult to rely on a probabilistic model. Additionally, even if a decision maker had a reasonable model of review scores, these probabilities may not represent meaningful probabilities to the funder, since review scales are arbitrary and likely do not map linearly to the utility of selecting a candidate. Finally, the future success of candidates in the settings we consider may be inherently hard to predict. In the context of scientific peer review, many works find that it is difficult to predict future citations and peer review scores are poor predictors [SWL+22, CL21, WDH+24]. In the related area of college admissions, recent work demonstrated that predictive models used to rank candidates are highly unstable with respect to their training data, resulting in arbitrariness in the ranking from any single model [LHZ+24]. For these reasons, we assume that the funder does not have a reliable probabilistic model of their data.
>
>     Thus, the funder operates in a setting of “Knightian uncertainty” where they estimate intervals that convey information about the relative quality of proposals but are not trusted as probabilistic estimates. We answer the question—how should they make decisions given this interval ordering? Given that any ordering consistent with the intervals is consistent with the decision maker’s information, taking the worst case over this set of orderings is a natural objective to optimize over their uncertainty.
>
>     We will expand on this motivation in the final version of the paper.
>
>     Additionally, one interesting direction along the lines you propose would be a smoothed analysis, which combines the average and worst case model. We conjecture that our MERIT algorithm would continue to be near optimal for that as well, but  leave a formal investigation to future work.
>
> 2. **Monotonicity vs. ex-ante optimality:** We provide a simple algorithm to enforce monotonicity with MERIT in Appendix D, which allows for violation of ex-ante optimality, thereby prioritizing the monotonicity property. We agree that monotonicity is a natural property. Interestingly, our analysis shows that existing randomized mechanisms of the Swiss NSF and randomize-above-threshold (which do not satisfy ex-ante optimality) also violate monotonicity.
>
> 3. **Results in Table 1:** Since there is no ground truth, we cannot say which algorithm is “better” at selecting high quality candidates based on this real world data. Therefore, our results in table 1 are meant to provide an interesting qualitative comparison of applying the Swiss NSF’s method and MERIT in real settings. The biggest takeaway in qualitative differences is that MERIT does in practice assign variable probabilities to different applicants (allowing for more nuanced lotteries), while Swiss NSF enforces a uniform lottery.
>
> ## References
>
> [Sha22] Nihar B Shah. An overview of challenges, experiments, and computational solutions in peer review, June 2022.
>
> [BGK05] Lyle Brenner, Dale Griffin, and Derek J Koehler. Modeling patterns of probability calibration with random support theory: Diagnosing case-based judgment. Organizational Behavior and Human Decision Processes, 97(1):64–81, 2005.
>
> [CL21]  Corinna Cortes and Neil D Lawrence. Inconsistency in conference peer review: Revisiting the 2014 NeurIPS experiment. arXiv Preprint arXiv:2109.09774, 2021.
>
> [WDH+24] Aidan S Weitzner, Matthew Davis, Andrew H Han, Olivia O Liu, Anuj B Patel, Brian D Sites, and Steven P Cohen. How predictive is peer review for gauging impact? The association between reviewer rating scores, publication status, and article impact measured by citations in a pain subspecialty journal. Regional Anesthesia & Pain Medicine, 2024.
>
> [LHZ+24] Jinsook Lee, Emma Harvey, Joyce Zhou, Nikhil Garg, Thorsten Joachims, and Rene F Kizilcec. Algorithms for college admissions decision support: Impacts of policy change and inherent variability. arXiv preprint arXiv:2407.11199, 2024.

---

> > ### Comment · Reviewer_BoxL · 2025-08-06
> > **Thank you for your response**
> >
> > Thank you for your clarification on ex-ante optimality. I don't have further questions. I keep my current rating.

---

### Decision · Program_Chairs · 2025-09-17

**Decision:**

Accept (spotlight)

**Comment:**

his paper proposes a principled and computationally efficient method for randomized selection under uncertainty, with applications to peer review, admissions, and funding decisions. The approach optimizes a maximin objective over interval uncertainty and satisfies ex-ante optimality and ex-post validity. It combines strong theoretical guarantees with a scalable algorithm and empirical evaluation.

The reviewers were consistently positive, highlighting the paper’s novelty, clarity, and practical relevance. The rebuttal addressed remaining concerns effectively, including motivation, modeling choices, and connections to prior work.